# Identification of COVID-19 spread mechanisms based on first-wave data, simulation models, and evolutionary algorithms

Vladimir Stanovov[1]*, Stanko Grabljevec[2], Shakhnaz Akhmedova[1], Eugene Semenkin[1], Radovan Stojanović[3], Črtomir Rozman[4], Andrej Škraba[5]

**1** Siberian Institute of Applied System Analysis Named After A.N. Antamoshkin, Reshetnev Siberian State University of Science and Technology, Krasnoyarsk, Krasnoyarsk Krai, Russian Federation, **2** Department of Anesthesiology and Perioperative Intensive Care, University Medical Centre Ljubljana, Ljubljana, Slovenia, **3** Department of Electrical Engineering and Computer Technology, Faculty of Electrical Engineering, University of Montenegro, Podgorica, Montenegro, **4** Department of Agricultural Economics, Faculty of Agriculture and Life Sciences, University of Maribor, Hoče, Slovenia, **5** Department of Informatics, Cybernetics & Decision Support Systems Laboratory, Faculty of Organizational Sciences, University of Maribor, Kranj, Slovenia

* vladimirstanovov@yandex.ru

**Data Availability Statement:** All relevant data are within the manuscript and within the Github

## Abstract

### Background

The COVID-19 epidemic has shown that efficient prediction models are required, and the well-known SI, SIR, and SEIR models are not always capable of capturing the real dynamics. Modified models with novel structures could help identify unknown mechanisms of COVID-19 spread.

### Objective

Our objective is to provide additional insights into the COVID-19 spread mechanisms based on different models' parameterization which was performed using evolutionary algorithms and the first-wave data.

### Methods

Data from the Our World in Data COVID-19 database was analysed, and several models—SI, SIR, SEIR, SEIUR, and Bass diffusion—and their variations were considered for the first wave of the COVID-19 pandemic. The models' parameters were tuned with differential evolution optimization method L-SHADE to find the best fit. The algorithm for the automatic identification of the first wave was developed, and the differential evolution was applied to model parameterization. The reproduction rates (R0) for the first wave were calculated for 61 countries based on the best fits.

repository: https://github.com/VladimirStanovov/OWID-COVID-19-Analysis.

**Funding:** This work was supported by the Ministry of Science and Higher Education of the Russian Federation within limits of state contract № FEFE-2020-0013 and by the Slovenian Research Agency (ARRS) (programs No.: UNI-MB-0586-P5-0018, No.: BI-RU/19-20-034, No. BI-ME/18-20-009). The funders had no role in study design, data collection and analysis, decision to publish, or preparation of the manuscript.

**Competing interests:** The authors have declared that no competing interests exist.

**Abbreviations:** BD, Bass diffusion model; DE, differential evolution; DRC, Democratic Republic of the Congo; GDP, gross domestic product; eSEIR, exponential susceptible-exposed-infected-recovered; eSEIUR, exponential susceptible-exposed-infected-unreported-recovered; eSI, exponential susceptible-infected; eSIR, exponential susceptible-infected-recovered; SEIR, susceptible-exposed-infected-recovered; SEIUR, susceptible-exposed-infected-unreported-recovered; SI, susceptible-infected; SIR, susceptible-infected-recovered; UK, United Kingdom.

## Results

The performed experiments showed that the Bass diffusion model-based modification could be superior compared to SI, SIR, SEIR and SEIUR due to the component responsible for spread from an external factor, which is not directly dependent on contact with infected individuals. The developed modified models containing this component were shown to perform better when fitting to the first-wave cumulative infections curve. In particular, the modified SEIR model was better fitted to the real-world data than the classical SEIR in 43 cases out of 61, based on Mann–Whitney U tests; the Bass diffusion model was better than SI for 57 countries. This showed the limitation of the classical models and indicated ways to improve them.

## Conclusions

By using the modified models, the mechanism of infection spread, which is not directly dependent on contacts, was identified, which significantly influences the dynamics of the spread of COVID-19.

## Introduction

The COVID-19 pandemic [1] has demonstrated the need for efficient infection-spread prediction models that are capable of capturing the dynamics of the process in different populations. Some of the well-known models often used for epidemiological dynamics are the susceptible–infected (SI) [2], susceptible–infected–recovered (SIR) [3], and the susceptible–exposed–infected–recovered (SEIR) models [4, 5]. These models are often described with several differential equations, controlled by population-based and infection-based parameters.

The mentioned models, especially the SEIR model, are often used in epidemiological studies due to their flexibility, simplicity, and transparency. Despite the popularity and simplicity of the basic models and their ability to give reasonable predictions, there are numerous modifications to SI, SIR and SEIR that allow for the consideration of additional factors that influence the dynamics of infection spread [6, 7]. The SEIR model is often modified, for example, at the SEIRU model [8] in addition to the reported infected cases, the unreported infected cases are also considered. In addition, this study introduces additional latency periods into the dynamics of the COVID-19 pandemic. In [9], a similar modification was applied—i.e., the reported and unreported infected cases were considered; however, following an early study on COVID-19 in [1], a metapopulation model was applied, which also considered human mobility networks. Obviously, the consideration of asymptomatic cases requires the estimation of their ratio, as well as their possible influence on the number of new infections [10, 11]. In [12], the extended SEIR model was applied, where detectable, undetectable, and isolated cases were considered, and an estimation of their number and influence were provided.

Based on these studies, it is clear that the proposed model modifications often introduce additional states describing real processes within the population; however, some of these models are often complicated and difficult to tune. The Bass diffusion model (BD) [13] is rarely used in epidemiological studies [14, 15], yet it has gained more recent interest given its simplicity and ability to build S-shaped curves. In [15], the BD model was also identified as one of most suitable for real data fitting. In this study, however, the discrete form of the BD model was applied, while in the present study, the differential equations form was considered, which

is typically used for modelling. The original BD model is most often used to show the spread of specific products in the market, but it could also be adapted to describing the spread of some infections. One of the features of interest in the BD model is that spreading may begin even if there are no product buyers/infected people at the outset of the study period.

The focus of the present study is the analysis of different types of models that could be applied to the description of real infection dynamics observed in different countries around the world for COVID-19 spread. The goal of our study was to compare the efficiency of different epidemic models in describing the real dynamics, and not to forecast the future spread. In particular, the mentioned models are fitted for each country included in the study, and conclusions are made about the efficiency of the models in these settings. One should note that the parameters in the presented models were fixed, which might be argued due to the facts, that countermeasures were forced in most countries, however, at least the first part of the first wave took majority of us by surprise. By fixed model parameters, one could extract possible novel epidemic spread mechanisms, otherwise, it would be possible to consider time varying infectivity factors which, however, might lead to model overfitting and losing the important information about the process. The modifications inspired by the Bass diffusion model were incorporated into the SIR, SEIR, and SEIUR models, and an additional set of experiments was performed. Based on the results, the importance of the modification is discussed, as well as its role in explaining the ongoing observed epidemiological reality.

The rest of the paper is organised as follows. First, the classical models are described, and their similarities and differences are highlighted. After this, the modified models are proposed, followed by the presentation of the data preparation method. Next, the modified models are described and the results of the experiments are given. Finally, the conclusions are provided.

## Methods

### Basic epidemiological models

To approximate the real data, containing the number of infected for every country, five basic models were considered—the SI, SIR, SEIR, SEIUR, and BD models.

The simplest SI model is described by a pair of differential equations:

$$\begin{cases} \dfrac{dS}{dt} = -\beta \dfrac{IS}{N} \\ \dfrac{dI}{dt} = \beta \dfrac{IS}{N} \end{cases}$$

where $\beta$ is the parameter controlling the flow from susceptible to infected; S and I are the number of susceptible and infected people, respectively; and N is the maximum number of susceptible. Note that if the number of infected is initially set to zero, there will be no dynamics, as the flow rate depends on the product of two states. Fig 1 shows the cumulative number of infected with $\beta = 0.1$, initial number of susceptible $S_0 = 100000$, and initial infected $I_0 = 1$, exercising S-shaped growth.

The SIR model can be described with the following equations:

$$\begin{cases} \dfrac{dS}{dt} = -\beta \dfrac{IS}{N} \\ \dfrac{dI}{dt} = \beta \dfrac{IS}{N} - \gamma I \\ \dfrac{dR}{dt} = \gamma I \end{cases}$$

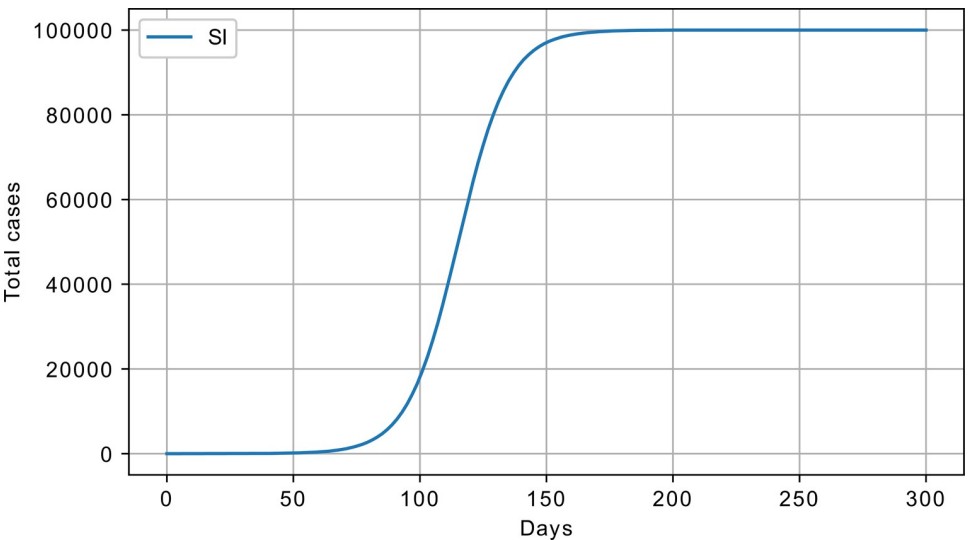

**Fig 1. SI model response, cumulative number of infected; $\beta = 0.1$, $S_0 = 100000$, and $I_0 = 1$.**

where $\beta$ and $\gamma$ are the model parameters, and $S$, $I$ and $R$ are the number of susceptible, infected, and recovered, respectively. The $\gamma$ parameter controls the exponential flow rate from infected to recovered. Alternatively, $\frac{1}{\gamma}$ can be seen as the number of days, $D$, required to get from the infected to recovered state.

A sample response of the SIR model is shown in Fig 2, with $S_0 = 100000$, $I_0 = 1$, $\beta = 0.1$, and $\gamma = 0.01$.

The main difference between SI and SIR is that in the SI model, the number of infected never decreases, while in the SIR model, the infected transition to the recovered state after certain period of time.

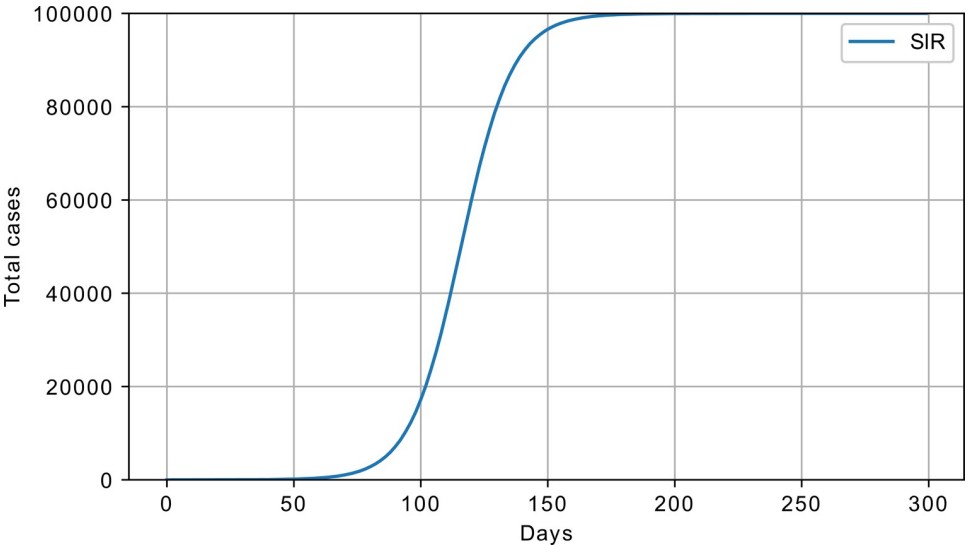

**Fig 2. SIR model response, cumulative number of infected; $\beta = 0.1$, $\gamma = 0.01$, $S_0 = 100000$, and $I_0 = 1$.**

The SEIR [16] model equations are shown below:

$$\begin{cases} \dfrac{dS}{dt} = -\beta I \dfrac{S}{N} \\[2mm] \dfrac{dE}{dt} = \beta I \dfrac{S}{N} - \sigma E \\[2mm] \dfrac{dI}{dt} = \sigma E - \gamma I \\[2mm] \dfrac{dR}{dt} = \gamma I \end{cases}$$

where $\beta$, $\sigma$, and $\gamma$ are the model parameters, and $S$, $E$, $I$, and $R$ are the number of susceptible, exposed infected, and recovered. The value $\frac{1}{\sigma}$ can be seen as the number of days, $Z$, required to move from the exposed to infected states [17]. An example response of the SEIR model is shown in Fig 3, with $S_0 = 100000$, $I_0 = 1$, $\beta = 1$, $\sigma = 0.01$, and $\gamma = 0.0005$.

In addition to the SEIR model, its extended version, the SEIUR model [8, 18, 19], was considered, which also included unreported cases:

$$\begin{cases} \dfrac{dS}{dt} = -\beta I \dfrac{S}{N} - \mu \beta U \dfrac{S}{N} \\[2mm] \dfrac{dE}{dt} = \beta I \dfrac{S}{N} + \mu \beta U \dfrac{S}{N} - \sigma E \\[2mm] \dfrac{dI}{dt} = \lambda \sigma E - \gamma I \\[2mm] \dfrac{dU}{dt} = (1 - \lambda)\sigma E - \gamma U \\[2mm] \dfrac{dR}{dt} = \gamma I + \gamma U \end{cases}$$

where $\beta$, $\sigma$, $\gamma$, and $\lambda$ are the model parameters, and $S$, $E$, $I$, $U$, and $R$ are the number of susceptible, exposed, infected (reported), unreported infected, and recovered. The two additional parameters of this model are the $\lambda$, responsible for the ratio of reported infected, and $\mu$,

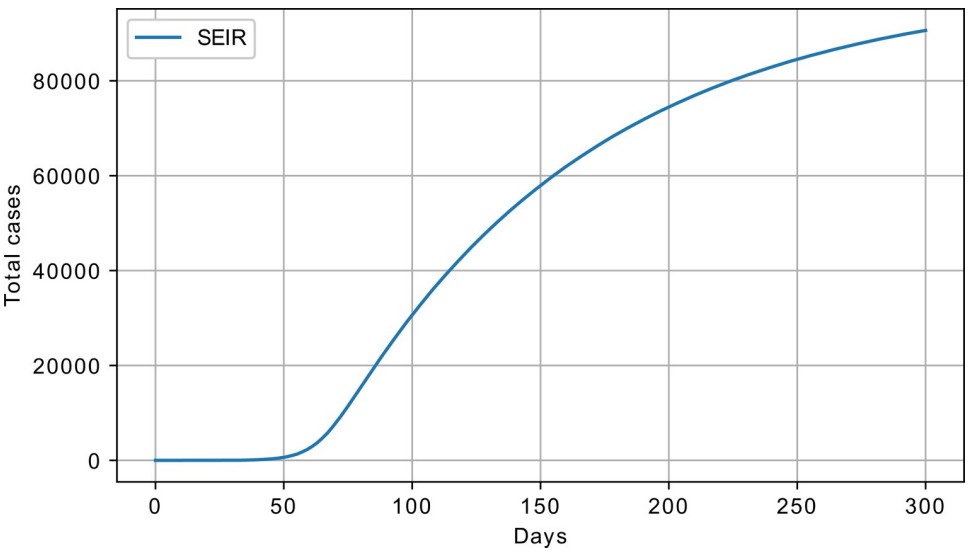

**Fig 3. SEIR model response, cumulative number of infected; $\beta = 1$, $\sigma = 0.01$, $\gamma = 0.0005$, $S_0 = 100000$, and $I_0 = 1$.**

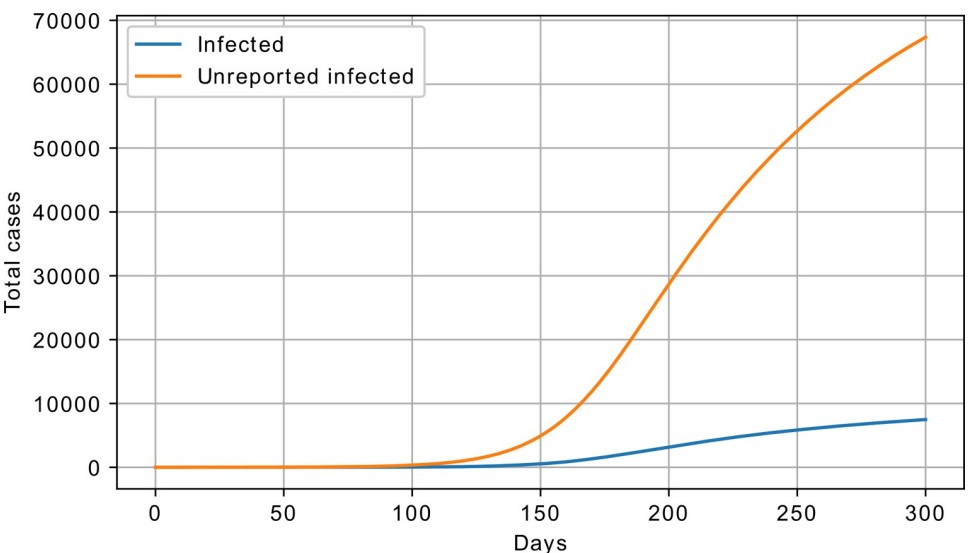

**Fig 4. SEIUR model response, cumulative number of infected; $\beta = 2$, $\sigma = 0.025$, $\gamma = 0.07$, $\lambda = 0.1$, $\mu = 0.3$, $S_0 = 100000$, and $I_0 = 1$.**

responsible for the influence ratio of the unreported infected. A sample response of the SEIUR model is shown in Fig 4.

The original Bass diffusion model [13] can be described as follows:

$$f(T) = (p + qF(T))(1 - F(T))$$

$$F(T) = \int_0^T f(t)dt$$

The $p$ and $q$ are the model parameters, the $p$ value is the coefficient of innovation, and $q$ is the coefficient of imitation. The $F(T)$ is defined as the cumulative of $f(T)$, which is the purchase likelihood function. The initial condition $F(0) = 0$ could be declared as 0 due to the innovation part of the model. Here the analogy is used, in particular the number of purchases is analogous to the number of infected, and the number of potential customers (determined by $1 - F[T]$) is analogous to the number of susceptible. Unlike in the SI model, here it is possible to become infected without interaction with the infected—e.g., being infected by external sources, such as animals. The graphical representation of this equation in PowerSim software system [20] is shown in Fig 5, and Table 1 contains the parameters.

An alternative way of describing the same model is as follows:

$$\frac{dN_t}{dt} = p(M - N_t) + \frac{q}{M}N_t(M - N_t)$$

where $N_t$ is the current total number of purchases (infected) and $M$ is the total number of purchases (total susceptible). These could also be interpreted as current number of infected and total number of susceptible. The graphical representation of this equation in PowerSim is shown in Fig 6, and Table 2 contains the parameters.

An example response of the BD model is shown in Fig 7. Here, as well as for all the other models, the cumulative number of infected is shown.

All presented models were further tested against the real data over a set of countries.

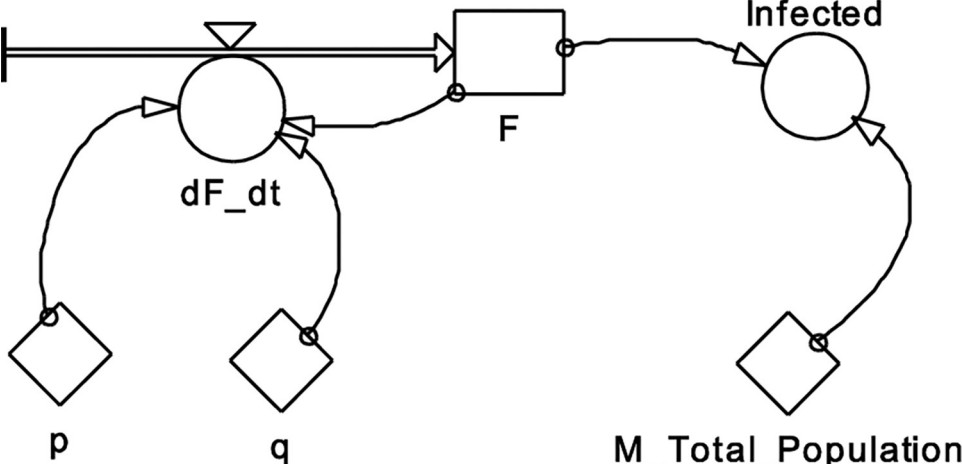

**Fig 5. BD model, first variant.**

**Modified models.**  In addition to the models described above, the modified models were applied, with the modification inspired by the Bass diffusion model and supported by the experimental results. The exact reasons for such modifications are described in the following sections.

The proposed modification incorporates the exponential outflow from susceptible to infected with $\theta$ parameter. This parameter is responsible for the rate of progression from susceptible to infectious without the interaction between infectious and susceptible. The modified SI model, also referred to as exponential SI (eSI), can be described with the following equations:

$$
\begin{cases}
\dfrac{dS}{dt} = -\beta \dfrac{IS}{N} - \theta S \\[2mm]
\dfrac{dI}{dt} = \beta \dfrac{IS}{N} + \theta S
\end{cases}
$$

where $\theta$ is responsible for the external factor, which influences the infections dynamics. An example response of the modified SI is shown in Fig 8.

The modified SI model has two parameters, which play the same roles as in the BD model. In fact, these two models are identical: one of the parameters controls the exponential outflow from the susceptible, while the other controls interactions between the susceptible and infected. With the same parameter values, the BD and modified SI demonstrate the same dynamics. Going forward in this paper, eSI is referred to as the BD model.

**Table 1. Parameters for BD model in Fig 5.**

| init | F = 0.1 |
|---|---|
| flow | F = +dt*dF_dt |
| aux | dF_dt = (p+q*F)*(1-F) |
| aux | Infected = M_Total_Population*F |
| const | M_Total_Population = 100000 |
| const | p = 0.0001 |
| const | q = 0.05 |

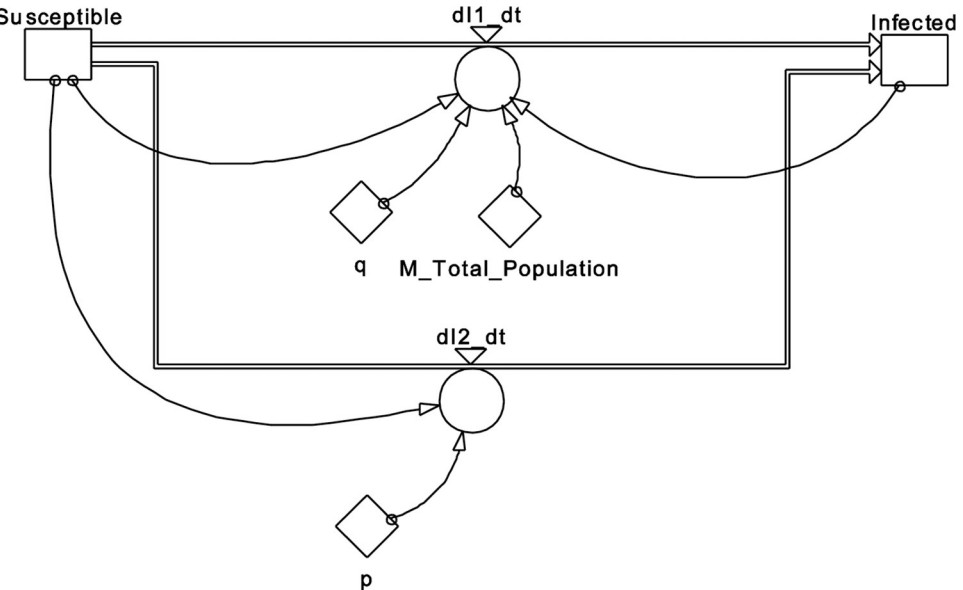

**Fig 6. BD model, second variant.**

The modified SIR (exponential SIR, eSIR) model is described as follows:

$$
\begin{cases}
\dfrac{dS}{dt} = -\beta \dfrac{IS}{N} - \theta S \\[2mm]
\dfrac{dI}{dt} = \beta \dfrac{IS}{N} - \gamma I + \theta S \\[2mm]
\dfrac{dR}{dt} = \gamma I
\end{cases}
$$

An example response of the original and modified SIR models is shown in Fig 9.

For the modified SIR model, the following parameters were used: $\beta = 0.1$, $\gamma = 0.01$, and $\theta = 1 \cdot 10^{-4}$; for the original SIR, $\beta = 0.06$ and $\gamma = 0.01$ were used. These parameters were chosen so that the two curves would better match each other. As one may observe, the SIR model begins with a lower rate and stabilises faster, whereas the initial rate of the modified SIR is higher and takes longer to stabilise at the end. This slower approach to the limit value was also observed in the real data, which is one of the desired model properties.

**Table 2. Parameters for BD model in Fig 6.**

| | |
|---|---|
| init | Infected = 0 |
| flow | Infected = +dt*dI1_dt+dt*dI2_dt |
| init | Susceptible = 100000 |
| flow | Susceptible = -dt*dI1_dt-dt*dI2_dt |
| aux | dI1_dt = q*(Susceptible/M_Total_Population)*Infected |
| aux | dI2_dt = p*Susceptible |
| const | M_Total_Population = 1e6 |
| doc | M_Total_Population = The size of the total population. |
| const | p = 0.0001 |
| const | q = 0.05 |

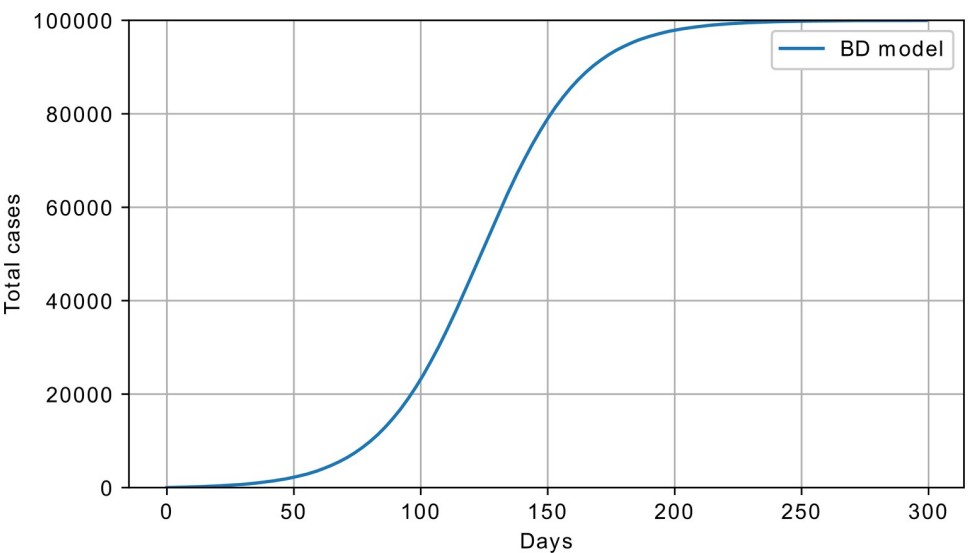

**Fig 7. Bass diffusion model response;** $p = 10^{-4}$, $q = 0.05$, $S_0 = 100000$, **and** $I_0 = 0$.

The SEIR model was changed to include the $\theta$ parameter. The modified SEIR (exponential SEIR, eSEIR) model equations are shown below:

$$
\begin{cases}
\dfrac{dS}{dt} = -\beta I \dfrac{S}{N} - \theta S \\[2mm]
\dfrac{dE}{dt} = \beta I \dfrac{S}{N} - \sigma E + \theta S \\[2mm]
\dfrac{dI}{dt} = \sigma E - \gamma I \\[2mm]
\dfrac{dR}{dt} = \gamma I
\end{cases}
$$

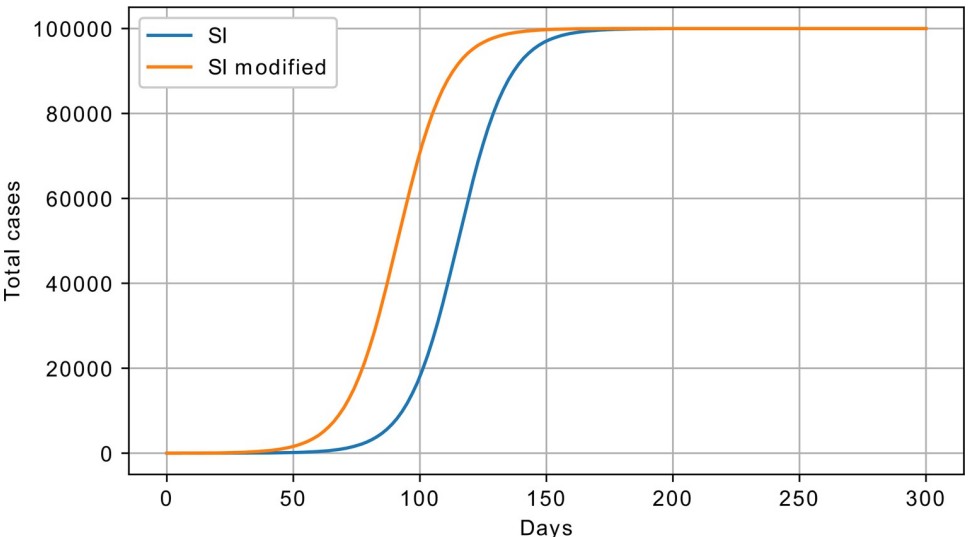

**Fig 8. SI vs eSI model response;** $\beta = 0.1$, $\theta = 1 \cdot 10^{-5}$, $S_0 = 100000$, **and** $I_0 = 1$.

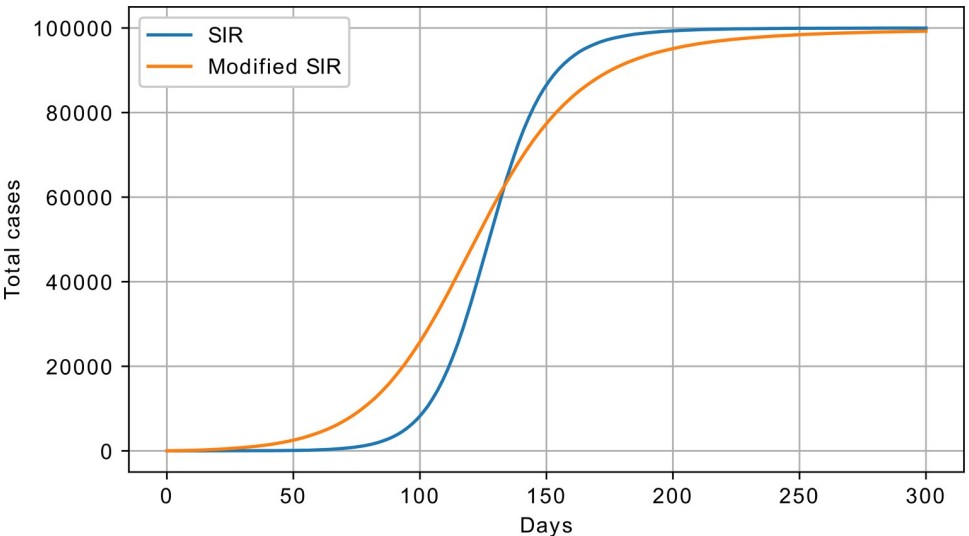

**Fig 9. SIR vs. eSIR model response; $S_0$ = 100000 and $I_0$ = 1.**

where $\theta$ plays the same role as in the modified SIR model. A sample response of the modified eSEIR model is shown in Fig 10.

For the modified SEIR model, the following parameters were used: $\beta = 1$, $\sigma = 0.006$, $\gamma = 0.0005$, and $\theta = 0.001$; for the original SEIR, $\beta = 1$, $\sigma = 0.01$, and $\gamma = 0.0005$ were used. Again, these parameters were chosen so that the two curves would better match each other. In both the modified SIR and SEIR models, if the $\theta$ parameter is set to zero, the models turn into their unmodified versions. As in the case of the SIR model, the SEIR model begins slowly and takes the limit value sooner, while the modified SEIR model begins its growth earlier and at a greater rate, but it takes more time to achieve the limit value. This property of reaching the limit value slower was also observed in the real data.

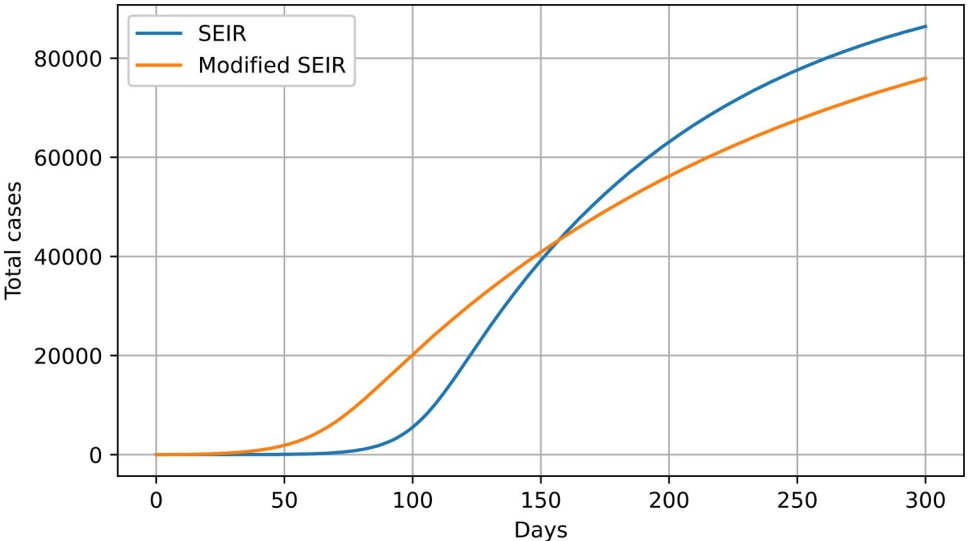

**Fig 10. SEIR vs. eSEIR model response; $S_0$ = 100000 and $I_0$ = 1.**

The SEIUR model was also modified to include the $\theta$ parameter, and the equations are given below:

$$\begin{cases} \dfrac{dS}{dt} = -\beta I \dfrac{S}{N} - \mu \beta U \dfrac{S}{N} - \theta S \\[2mm] \dfrac{dE}{dt} = \beta I \dfrac{S}{N} + \mu \beta U \dfrac{S}{N} - \sigma E + \theta S \\[2mm] \dfrac{dI}{dt} = \lambda \sigma E - \gamma I \\[2mm] \dfrac{dU}{dt} = (1-\lambda)\sigma E - \gamma U \\[2mm] \dfrac{dR}{dt} = \gamma I + \gamma U \end{cases}$$

A comparison of the SEIUR and its modified version (exponential SEIUR, eSEIUR) is shown in Fig 11.

In Fig 11, the following parameters were used for the SEIUR model: $\beta = 2$, $\sigma = 0.025$, $\gamma = 0.07$, $\theta = 0$, $\lambda = 0.1$, and $\mu = 0.3$. For the eSEIUR, the following parameters were set: $\beta = 2$, $\sigma = 0.025$, $\gamma = 0.07$, $\theta = 0.001$ $\lambda = 0.1$, and $\mu = 0.3$. The next section contains a description of the data preparation.

**Data preparation.** The dataset considered in this study was taken from the GitHub repository using the Our World in Data database containing the COVID-19 spread dynamics in every country and region [21]. The dataset contained the number of infected persons daily since the beginning of observations, as well as the cumulative number of infected. In addition, some other data is provided for each country, such as population, GDP, etc.

In this study, the cumulative number of infected was considered—i.e., the models presented above were used to fit to the actual cumulative infection curves. The dataset contained information about several waves of infection for many countries, however for the purpose of our study only the first wave—the initial spread of the infection—was considered [22]. The data from the first wave were considered to extract the basic characteristics of the process, which would be more difficult to extract in second and consecutive waves. Due to different start

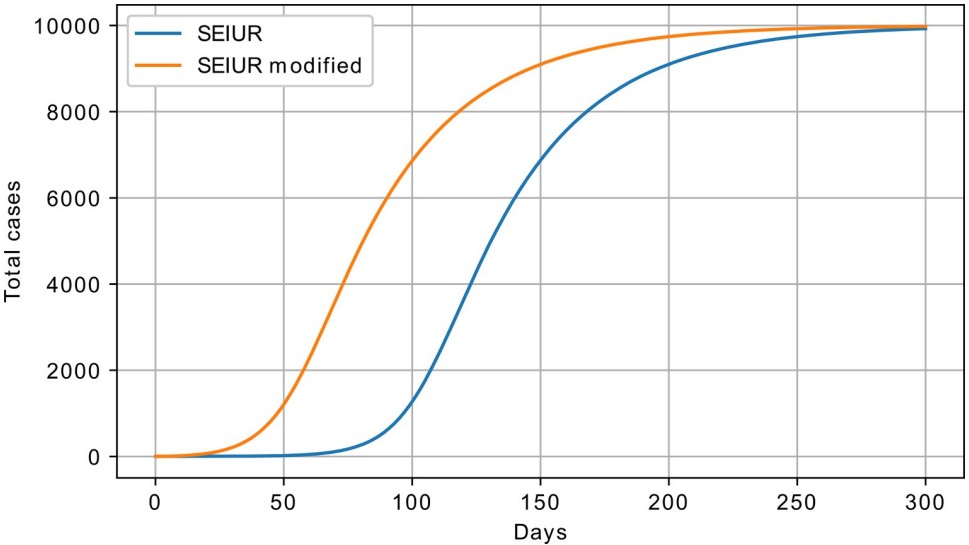

**Fig 11. SEIUR vs. eSEIUR model response; $S_0 = 100000$ and $I_0 = 1$.**

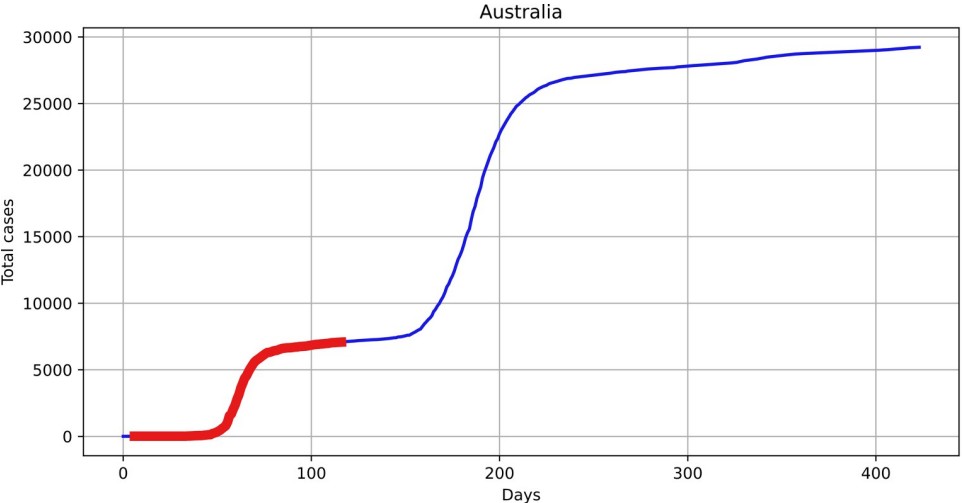

**Fig 12. Example of two waves of infections; the first wave is highlighted.**

times of the first wave in different countries, it was necessary to cut several first measurements. To identify the starting points, $St_{min}$, the following heuristic rule was used: if for $\tau = 3$ time steps before $t$, the exponentially smoothed (with parameter $\varepsilon = 0.2$) total number of infected was increasing by at least 1 case, then $St_{min} = t$.

The algorithm for detecting the end of the first wave is more complex and considers several steps. First, the number of infected per day was considered, and according to the observations, the first wave ended at the point where the number of infected per day dropped closer to zero after a relatively high peak (see Fig 12).

Fig 12 shows that it should be possible to identify the end of the first wave by determining the position of the first minimum after the first maximum of daily infections. However, it could be challenging to do so, as there were many oscillations in the number of infections per day. As the provided dataset contained already smoothed number of infected per day, this data was used in the experiments. To identify the minimum, the estimation of the derivative of the number of infected per day was calculated with finite differences:

$$\hat{I}_t = I_{t+1} - I_t$$

where $t = 1 \ldots T-1$. The approximate derivative was further processed using an exponential smoothing approach:

$$\hat{\hat{I}}_{t+1} = \varepsilon \cdot \hat{I}_{t+1} + (1 - \varepsilon) \cdot \hat{I}_t$$

where $t = 1 \ldots T-1$, $\varepsilon = 0.02$. The resulting values were analysed to detect the first wave following these three rules:

1. $\hat{I}_t$ should be positive for at least $wp \cdot T$ steps, after this

2. $\hat{I}_t$ should be negative for at least $wp \cdot T$, and then

3. $\hat{I}_t$ should switch from negative to positive value.

The highlighted area in Fig 12 shows the results of automatic detection, and Fig 13 shows the smoothed number of infections per day and the smoothed derivative.

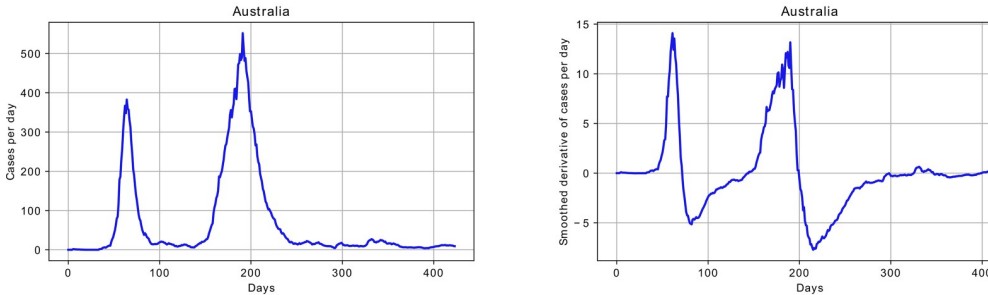

**Fig 13. Example of number of cases per day and calculated smoothed derivative.**

For the case in Figs 12 and 13, the *wp* parameter was set to 0.07, and the detected end of the first wave and beginning of the second was identified at day 145.

The two main parameters controlling the proposed heuristic for determining the first wave are *τ* and *wp*. To demonstrate the influence of these values, Fig 14 shows the possible scenarios of determining the first wave with varied parameter values.

The goal was to analyse the first wave of the infection, but because several countries were still far from the inflection point, they were filtered out. In addition, countries with populations of less than 1 million people were not considered. Several countries were also removed because either very few cases were reported relative to the total population, or they were special cases, like China, where a very small fraction of new infections were reported after a long period of time. Other countries that were excluded due to peculiarities in the data were Denmark, Brazil, Uganda, Vietnam, Poland, and Northern Macedonia. For these countries, the first wave did not follow the S-shaped curve, and the first wave was not clearly distinguished from the second. This limitation was made due to possible biased dynamics for very small countries, such as islands or cities, which are significantly affected by their location, tourism, and other factors.

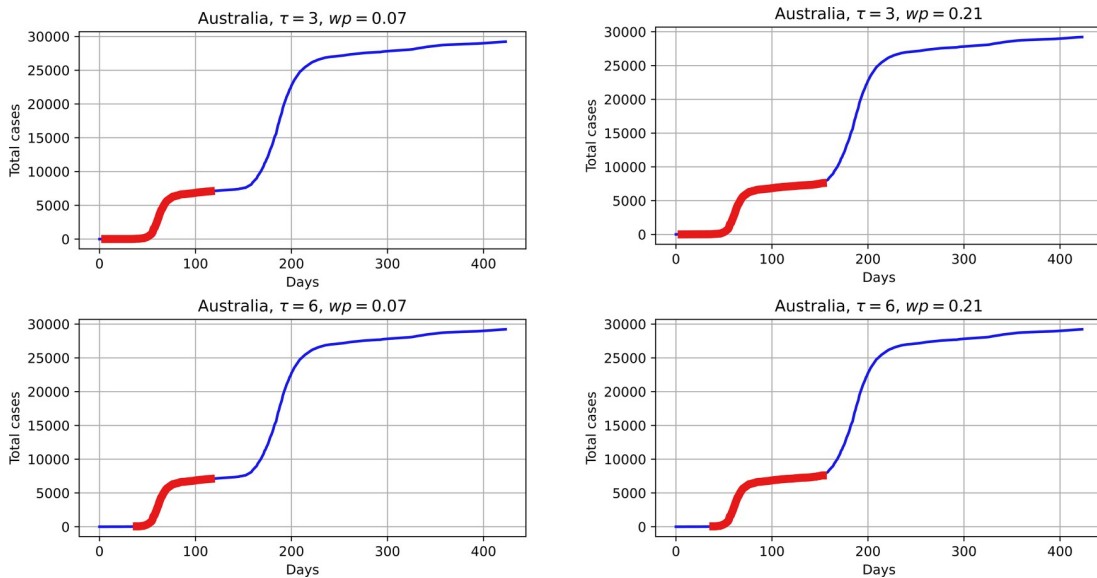

**Fig 14. Influence of first wave search heuristic parameters on the result.**

**Differential evolution.** There are several known attempts to apply computational intelligence methods, such as neural networks, for prediction of COVID-19 epidemic trends [23]. In our study, to find the parameter values for the presented models, the differential evolution (DE) algorithm was applied [24]. In particular, a well-known adaptive version, L-SHADE [25], is implemented and modified to find the parameter values of the presented models.

The DE algorithm performed the search for optimal solutions in $R^D$, where D is the dimensionality. DE begins with initialising a set of vectors: $x_{i,j} = Random(L_j, U_j)$, where $i = 1...NP$, $j = 1...D$, $L_j$, and $U_j$ are the lower and upper boundaries, and $NP$ is the population size. The fitness values $f(x_i)$—that is, the goal function values—were calculated for every individual.

After the initialisation step, the DE proceeded with the main algorithm loop, containing mutation, crossover, and selection steps. The mutation in DE is the main search operator, which combines the coordinates of several points to produce new ones. The mutation strategy used in L-SHADE is called *current-to-pbest/1* and generates new mutant vector, *v*, using the scaling factor, *F*, as follows:

$$v_j = x_i + F(x_{pbest} - x_i) + F(x_{r1} - x_{r2})$$

where *r1* and *r2* are the randomly chosen indexes from [1, NP], *pbest* is one of the *p*% fittest solutions, and *i* is the index of current vector. Indexes *i*, *r1*, *r2* and *pbest* are different from each other. The *r2* index is chosen from either the population or the external archive, composed of individuals replaced during selection.

The mutation step was followed by crossover, which combined the target vector $x_i$ and the mutant vector *v* to the produce trial vector *u* in the following way:

$$u_j = \begin{cases} v_j \; if \; Random(0, 1) < Cr \; or \; j = jrand \\ x_{i,j} \; otherwise \end{cases},$$

where $j = 1...D$, *Cr* is the crossover rate parameter and *jrand* is the randomly chosen integer from [1, D] needed to ensure that at least one component is taken from the mutant vector.

Once the trial vector was generated, its fitness $f(u)$ was calculated and compared to the fitness of the target vector in the selection procedure:

$$x_i^{G+1} = \begin{cases} u \; if \; f(u) < f(x_i^G) \\ x_i^G \; otherwise \end{cases},$$

where *G* is the current generation number—i.e., selection forms the new generation of individuals. The replaced parent is placed into the archive, and if the archive size reaches *NP*, then the random solution from the archive is replaced.

The parameter adaptation is one of the features of L-SHADE method. For every selection and mutation the *F* and *Cr* parameters are sampled from Cauchy and normal distributions with scale parameter equal to 0.1: $F = randc(M_{F,k}, 0.1)$, $Cr = randn(M_{Cr,k}, 0.1)$. The algorithm maintains a set of *H* memory cells each containing a pair of $(M_{F,k}, M_{Cr,k})$ values. One of the memory cells is updated at the end of each generation using weighted Lehmer mean of stored successful *F* and *Cr* values in $S_F$ and $S_{Cr}$, with weights being the fitness improvements$\Delta f = |f(u_j) - f(x_j)|$ in the selection step, stored in $S_{\Delta f}$:

$$mean_{wL} = \frac{\sum_{j=1}^{|S|} w_j S_j^2}{\sum_{j=1}^{|S|} w_j S_j},$$

where $w_j = \frac{S_{\Delta f_j}}{\sum_{k=1}^{|S|} S_{\Delta f_k}}$, $S$ is either $S_F$ or $S_{Cr}$. The memory cells are updated as follows:

$$M_{F,k}^{t+1} = mean_{wL,F}$$

$$M_{Cr,k}^{t+1} = mean_{wL,Cr}$$

where $t$ is the current iteration number, iterating from 1 to $H$. The L-SHADE algorithm also reduces the population size in the following way:

$$NP_{g+1} = round\left(\frac{NP_{min} - NP_{max}}{NP_{max}} NFE\right) + NP_{max},$$

where $NP_{min} = 4$, $NFE$ and $NFE_{max}$ are the current and total number of function evaluations available, $g$ is the current generation number. At the end of each generation the worst solutions are removed, if the population size changes.

In addition to the parameter adaptation, for the goals of current study the L-SHADE algorithm was modified by a relatively simple restart heuristic. If the best found solution does not change for $0.25 NFE_{max}$ function evaluations, then the whole population is randomized, except for the best solution. The idea of the modification was to allow the algorithm to escape local optima if it gets stuck. Note that the resulting L-SHADE-R method restarts with a smaller population size, depending on the used computational resource.

## Results

To compare the ability of the presented models to describe the real-world dynamics, the models were tuned to fit the available data regarding the first wave of the COVID-19 pandemic. To choose the appropriate parameter values for the SI, BD, SIR, SEIR, and SEIUR models, as well as the modified versions, the differential evolution algorithm was applied. The DE represents a universal zero-order continuous optimiser, capable of finding the global optimum even in the case of rough goal function terrain, as demonstrated in many benchmarks and real-world applications. All models were implemented in Python 3.7 with Odeint, which is a part of the SciPy library [26]. To perform experiments, the parallel implementation with OpenMPI was utilized, and the search for parameter values of each model on each country data was performed on a cluster of 12 computers with 8 cores each, connected via network.

In addition to the model parameters mentioned above, one additional parameter, $\alpha$, was considered, controlling the amount of susceptibility at the end of the first wave with respect to the total population: $S = \alpha P$, where $P$ is the total population of a country.

In the first series of experiments, the goal function, used as the fitness function in L-SHADE-R, was calculated as the relative error between the real data and the curve predicted by the model over all data points:

$$RE_c = \sqrt{\frac{\sum_{t=1}^{N} \left(CI_{c,t} - MI_t\right)^2}{\sum_{t=1}^{N} \left(CI_{c,t}\right)^2}}$$

where $N$ is the total number of cases in the first $T$ days, $c$ is the country index, $CI_{c,t}$ is the cumulative number of infected at day $t$ in country $c$, and $MI_t$ is the cumulative number of infections, all as predicted by the model at day $t$.

Preliminary, we have used several standard measures for fitness function such as Mean Absolute Percent Error (MAPE), Mean Absolute Error (MAE), Mean Squared Error (MSE).

However, the proposed relative error measure $RE_c$ was more stable in case of extreme values e.g. when the number of infected is close to 0.

The L-SHADE-R algorithm was executed with the following parameters:

1. Population size: 500 individuals,

2. Computational resource: 125000 evaluations,

3. Memory size $H = 5$, initial $M_{F,k} = 0.2$, $M_{Cr,k} = 0.8$,

4. Ratio of best for mutation $pbest = 0.2$,

5. Initial population generated within the following boundaries:

    a. $\alpha \in [0,0.1]$;

    b. $\beta \in [0,8]$;

    c. $\sigma \in \left[\frac{1}{42}, \frac{1}{2}\right]$;

    d. $\gamma \in \left[\frac{1}{42}, \frac{1}{2}\right]$;

    e. $\theta \in [0,0.1]$;

    f. $\lambda \in [0.0001,0.5]$;

    g. $\mu \in [0,8]$.

The initial conditions of the models, i.e. state variables, were set according to the available data. First value of the Infected, that were recorded in official statistics, were used for initialization of the Infected state. Other states, such as exposed, recovered and unreported, were initialized to zero.

The ranges for $\sigma$ and $\gamma$ were chosen according to [27], where transmission from exposed to infected and from infected to recovered were determined between 3 and 14 days. However, the lower boundary for these parameters was expanded to 1/42, i.e. 42 days, to test if this gives additional performance gains. For $\beta$ the search range was set from 0 to 8 according to preliminary experiments in literature, for example in [28] the $\beta$ parameter was set to from 1.5 to 3. Larger range was used to allow the search algorithm discover better solutions, so that post-processing would show the most preferable range to which parameter values fall into.

Different models had different dimensionalities, as the number of parameters varied. For SI there were two variables, for BD there were three variables, for SIR and SEIR there were three and four variables, and for eSIR and eSEIR there were four and five variables, respectively. Finally, SEIUR and eSEIUR had six and seven variables. The best parameter values for every model and every country were saved at the end of each algorithm run, and there were 25 independent runs performed for every model and every country. Table 3 contains the average relative error rates and their standard deviations for all countries and models, as well as the number of cases when a particular model was the best compared to others. In addition to standard settings, an experiment with extended range for eSEIUR, denoted with eSEIUR$^{\text{range}}$ was performed, with the following settings:

1. $\alpha \in [0,0.5]$;

2. $\beta \in [0,12]$;

3. $\sigma \in \left[\frac{1}{56}, \frac{1}{2}\right]$;

4. $\gamma \in \left[\frac{1}{56}, \frac{1}{2}\right]$;

**Table 3. Average relative error rates and their standard deviations for all countries and models.**

| Country | SI | BD | SIR | eSIR | SEIR | eSEIR | SEIUR | eSEIUR | eSEIUR$^{range}$ |
|---|---|---|---|---|---|---|---|---|---|
| Afghanistan | 3.809% | 4.315% | 1.299% | 0.006% | 0.013% | 0.008% | 0.012% | 0.016% | 0.024% |
| | ±15.267% | ±21.019% | ±6.284% | ±0% | ±0% | ±0% | ±0.001% | ±0.007% | ±0.015% |
| Armenia | 0.382% | 0.012% | 0.010% | 0.009% | 0.047% | 0.011% | 0.032% | 0.014% | 0.032% |
| | ±0% | ±0% | ±0% | ±0% | ±0% | ±0% | ±0.003% | ±0.003% | ±0.006% |
| Australia | 4.972% | 1.531% | 1.790% | 1.790% | 1.345% | 1.345% | 1.391% | 1.564% | 1.575% |
| | ±16.870% | ±0.013% | ±0% | ±0% | ±0% | ±0% | ±0.036% | ±0.161% | ±0.233% |
| Austria | 3.000% | 0.098% | 4.690% | 0.040% | 0.011% | 0.010% | 0.011% | 0.012% | 0.026% |
| | ±13.152% | ±0.018% | ±22.628% | ±0% | ±0% | ±0% | ±0% | ±0.002% | ±0.025% |
| Azerbaijan | 0.348% | 0.063% | 0.072% | 0.078% | 0.068% | 0.066% | 0.118% | 0.108% | 0.137% |
| | ±0% | ±0% | ±0% | ±0.003% | ±0.006% | ±0.001% | ±0.152% | ±0.008% | ±0.058% |
| Belarus | 1.031% | 0.119% | 0.305% | 0.023% | 0.063% | 0.015% | 0.055% | 0.015% | 0.020% |
| | ±0% | ±0.344% | ±0% | ±0% | ±0% | ±0% | ±0.004% | ±0.001% | ±0.004% |
| Belgium | 1.059% | 1.193% | 0.499% | 0.030% | 4.020% | 0.009% | 0.029% | 0.010% | 0.012% |
| | ±0% | ±5.451% | ±0.024% | ±0.001% | ±19.528% | ±0% | ±0.002% | ±0% | ±0.005% |
| Bolivia | 0.726% | 0.008% | 0.113% | 0.003% | 0.460% | 0.003% | 0.068% | 0.006% | 0.021% |
| | ±0.314% | ±0% | ±0% | ±0% | ±1.868% | ±0% | ±0.003% | ±0.003% | ±0.009% |
| Burkina Faso | 1.649% | 4.069% | 0.287% | 0.039% | 0.239% | 0.053% | 0.051% | 0.051% | 0.051% |
| | ±1.427% | ±19.582% | ±0% | ±0% | ±0.312% | ±0% | ±0% | ±0% | ±0% |
| Cambodia | 6.290% | 9.956% | 4.470% | 2.522% | 4.698% | 4.324% | 4.723% | 4.886% | 4.639% |
| | ±0.269% | ±20.100% | ±0% | ±0% | ±0% | ±0.911% | ±0.028% | ±0.226% | ±0% |
| Cameroon | 1.015% | 0.095% | 0.232% | 0.068% | 0.149% | 0.068% | 0.134% | 0.102% | 0.072% |
| | ±0% | ±0.002% | ±0.021% | ±0% | ±0% | ±0% | ±0.009% | ±0.146% | ±0.002% |
| Canada | 1.535% | 0.081% | 6.482% | 0.033% | 4.882% | 0.009% | 0.046% | 0.009% | 0.010% |
| | ±0% | ±0% | ±21.811% | ±0% | ±23.671% | ±0% | ±0.003% | ±0% | ±0.001% |
| Chile | 2.508% | 0.522% | 3.235% | 0.354% | 1.212% | 0.288% | 0.579% | 0.288% | 0.232% |
| | ±0% | ±0% | ±6.146% | ±0% | ±2.388% | ±0% | ±0.002% | ±0% | ±0.001% |
| Costa Rica | 1.386% | 0.220% | 0.617% | 0.084% | 0.416% | 0.056% | 0.311% | 0.061% | 0.048% |
| | ±0% | ±0.447% | ±0% | ±0% | ±0% | ±0% | ±0.004% | ±0% | ±0.001% |
| Cote d'Ivoire | 0.479% | 0.945% | 0.117% | 0.090% | 0.116% | 0.094% | 0.110% | 0.103% | 0.100% |
| | ±0% | ±4.032% | ±0% | ±0% | ±0% | ±0% | ±0.003% | ±0.007% | ±0.004% |
| Croatia | 0.434% | 8.238% | 1.099% | 0.027% | 0.024% | 0.019% | 0.023% | 0.022% | 0.018% |
| | ±0% | ±40.108% | ±5.171% | ±0% | ±0% | ±0.014% | ±0.003% | ±0.022% | ±0.001% |
| Cuba | 0.831% | 4.683% | 0.594% | 0.067% | 0.055% | 0.052% | 4.680% | 0.054% | 0.054% |
| | ±0% | ±19.625% | ±1.952% | ±0% | ±0% | ±0% | ±22.661% | ±0.001% | ±0.001% |
| Czechia | 0.672% | 5.920% | 1.037% | 0.044% | 9.239% | 0.013% | 0.914% | 0.015% | 0.016% |
| | ±0% | ±28.234% | ±3.997% | ±0% | ±45.152% | ±0% | ±4.386% | ±0.001% | ±0.002% |
| DRC | 0.285% | 0.060% | 0.039% | 0.026% | 0.023% | 0.014% | 0.016% | 0.016% | 0.016% |
| | ±0% | ±0% | ±0% | ±0% | ±0.016% | ±0% | ±0.001% | ±0.001% | ±0.003% |
| Egypt | 1.799% | 0.010% | 0.024% | 0.025% | 0.013% | 0.012% | 0.017% | 0.053% | 0.078% |
| | ±8.147% | ±0% | ±0% | ±0% | ±0% | ±0% | ±0.003% | ±0.022% | ±0.025% |
| El Salvador | 0.338% | 0.065% | 0.071% | 0.073% | 0.072% | 0.068% | 1.989% | 0.101% | 0.122% |
| | ±0% | ±0% | ±0% | ±0.004% | ±0.007% | ±0% | ±7.388% | ±0.007% | ±0.021% |
| Estonia | 0.704% | 0.060% | 0.172% | 0.033% | 0.091% | 0.042% | 0.081% | 0.056% | 0.047% |
| | ±0% | ±0.022% | ±0% | ±0% | ±0% | ±0.004% | ±0.007% | ±0.053% | ±0.002% |
| Finland | 1.352% | 0.021% | 0.367% | 0.073% | 9.906% | 0.009% | 0.077% | 0.361% | 0.016% |
| | ±0.461% | ±0% | ±0% | ±0.301% | ±33.652% | ±0% | ±0.005% | ±1.680% | ±0.004% |

(*Continued*)

**Table 3.** (Continued)

| Country | SI | BD | SIR | eSIR | SEIR | eSEIR | SEIUR | eSEIUR | eSEIUR^range |
|---|---|---|---|---|---|---|---|---|---|
| France | 1.142% | 2.337% | 0.854% | 0.810% | 0.953% | 0.765% | 0.993% | 0.836% | 0.863% |
| | ±0.301% | ±7.639% | ±0% | ±0.004% | ±0% | ±0% | ±0.166% | ±0.052% | ±0.080% |
| Germany | 0.317% | 0.170% | 0.104% | 0.088% | 0.021% | 0.021% | 0.134% | 0.042% | 0.055% |
| | ±0% | ±0% | ±0% | ±0% | ±0% | ±0% | ±0.506% | ±0.010% | ±0.037% |
| Greece | 0.564% | 0.085% | 9.928% | 0.041% | 0.036% | 0.033% | 0.041% | 0.035% | 0.035% |
| | ±0.437% | ±0% | ±4.176% | ±0% | ±0% | ±0% | ±0.030% | ±0.002% | ±0.001% |
| Haiti | 4.605% | 0.325% | 0.218% | 0.207% | 0.087% | 0.087% | 1.068% | 0.104% | 0.094% |
| | ±20.388% | ±0% | ±0% | ±0% | ±0% | ±0% | ±4.733% | ±0.009% | ±0.006% |
| Hungary | 0.807% | 0.165% | 0.159% | 0.032% | 0.032% | 0.014% | 0.028% | 0.016% | 0.015% |
| | ±0.038% | ±0.435% | ±0% | ±0% | ±0% | ±0% | ±0.003% | ±0.001% | ±0.001% |
| India | 3.691% | 0.334% | 0.767% | 0.013% | 0.286% | 0.005% | 0.269% | 0.006% | 0.007% |
| | ±11.113% | ±1.426% | ±0% | ±0% | ±0% | ±0% | ±0.012% | ±0% | ±0.002% |
| Iraq | 1.119% | 0.534% | 4.253% | 0.021% | 0.394% | 0.015% | 0.207% | 0.015% | 0.016% |
| | ±0% | ±1.577% | ±19.494% | ±0% | ±0.690% | ±0% | ±0.015% | ±0% | ±0.002% |
| Ireland | 34.795% | 0.036% | 0.094% | 0.015% | 4.696% | 0.016% | 0.040% | 0.018% | 0.031% |
| | ±165.776% | ±0% | ±0% | ±0% | ±22.780% | ±0% | ±0.004% | ±0.001% | ±0.020% |
| Israel | 0.609% | 0.136% | 0.177% | 0.082% | 0.026% | 0.026% | 0.027% | 0.031% | 0.031% |
| | ±0% | ±0% | ±0% | ±0% | ±0% | ±0% | ±0% | ±0.012% | ±0.007% |
| Italy | 1.157% | 21.866% | 0.599% | 0.029% | 0.019% | 0.007% | 0.017% | 0.007% | 0.008% |
| | ±0% | ±106.712% | ±0.291% | ±0% | ±0% | ±0% | ±0.001% | ±0% | ±0.001% |
| Japan | 1.876% | 1.441% | 0.342% | 0.342% | 0.167% | 0.213% | 0.205% | 0.297% | 0.398% |
| | ±7.348% | ±6.021% | ±0% | ±0% | ±0% | ±0.222% | ±0.035% | ±0.086% | ±0.111% |
| Latvia | 1.516% | 31.425% | 0.499% | 0.048% | 0.024% | 5.838% | 0.025% | 0.025% | 0.026% |
| | ±0.002% | ±134.403% | ±0% | ±0% | ±0% | ±28.481% | ±0.001% | ±0% | ±0.005% |
| Lebanon | 3.637% | 7.382% | 0.646% | 0.241% | 0.659% | 0.128% | 3.083% | 3.083% | 0.128% |
| | ±12.628% | ±35.513% | ±0% | ±0% | ±0.362% | ±0% | ±0% | ±0% | ±0% |
| Liberia | 0.317% | 0.233% | 0.271% | 0.269% | 0.252% | 0.252% | 0.312% | 0.320% | 0.353% |
| | ±0% | ±0.018% | ±0% | ±0% | ±0% | ±0% | ±0.261% | ±0.015% | ±0.017% |
| Lithuania | 1.293% | 0.373% | 0.870% | 3.394% | 0.025% | 26.968% | 0.027% | 0.025% | 0.025% |
| | ±0% | ±0.982% | ±1.907% | ±16.526% | ±0% | ±108.829% | ±0.010% | ±0% | ±0% |
| Mali | 0.745% | 0.052% | 0.055% | 0.065% | 0.088% | 0.063% | 0.163% | 0.090% | 0.095% |
| | ±0.009% | ±0% | ±0% | ±0.004% | ±0% | ±0% | ±0.022% | ±0.009% | ±0.012% |
| Mauritania | 0.661% | 4.179% | 0.190% | 0.046% | 0.037% | 0.033% | 0.036% | 0.035% | 0.035% |
| | ±0% | ±19.560% | ±0% | ±0% | ±0% | ±0% | ±0.001% | ±0.001% | ±0.001% |
| Mauritius | 0.591% | 0.578% | 1.368% | 0.293% | 1.228% | 0.306% | 0.571% | 0.334% | 0.376% |
| | ±0.032% | ±0% | ±3.962% | ±0% | ±3.219% | ±0% | ±0% | ±0.015% | ±0.054% |
| Netherlands | 1.146% | 0.944% | 4.820% | 0.018% | 1.856% | 0.011% | 0.058% | 0.012% | 0.016% |
| | ±0.001% | ±4.358% | ±21.075% | ±0% | ±8.783% | ±0% | ±0.003% | ±0% | ±0.006% |
| New Zealand | 0.122% | 0.013% | 3.489% | 4.006% | 0.015% | 0.005% | 0.015% | 0.006% | 0.007% |
| | ±0% | ±0% | ±17.049% | ±19.595% | ±0% | ±0% | ±0.021% | ±0% | ±0% |
| Norway | 1.570% | 0.078% | 2.143% | 0.026% | 0.060% | 0.023% | 0.052% | 0.025% | 0.026% |
| | ±3.419% | ±0% | ±8.968% | ±0% | ±0% | ±0% | ±0.003% | ±0.001% | ±0.002% |
| Oman | 26.340% | 2.054% | 4.082% | 0.084% | 0.062% | 0.059% | 0.070% | 0.060% | 0.192% |
| | ±108.907% | ±9.784% | ±19.582% | ±0.001% | ±0% | ±0.003% | ±0.004% | ±0.012% | ±0.045% |
| Pakistan | 1.847% | 2.013% | 0.134% | 0.041% | 0.159% | 0.033% | 0.149% | 0.068% | 0.101% |
| | ±6.357% | ±9.701% | ±0% | ±0.002% | ±0% | ±0.001% | ±0.006% | ±0.016% | ±0.029% |

(*Continued*)

**Table 3.** (Continued)

| Country | SI | BD | SIR | eSIR | SEIR | eSEIR | SEIUR | eSEIUR | eSEIUR[range] |
|---|---|---|---|---|---|---|---|---|---|
| Saudi Arabia | 4.048% | 0.046% | 2.352% | 0.016% | 0.163% | 0.017% | 0.145% | 0.017% | 0.019% |
| | ±10.426% | ±0.004% | ±6.030% | ±0% | ±0% | ±0.001% | ±0.008% | ±0% | ±0.002% |
| Serbia | 0.324% | 0.070% | 0.049% | 0.029% | 4.341% | 0.014% | 0.016% | 0.016% | 0.017% |
| | ±0% | ±0.009% | ±0% | ±0% | ±21.188% | ±0% | ±0% | ±0.001% | ±0.005% |
| Singapore | 1.006% | 0.408% | 3.490% | 0.293% | 0.134% | 0.134% | 0.135% | 0.135% | 0.106% |
| | ±0% | ±0% | ±10.129% | ±0% | ±0% | ±0% | ±0.001% | ±0.001% | ±0.010% |
| Slovakia | 0.306% | 0.884% | 0.151% | 0.148% | 0.148% | 0.138% | 0.224% | 0.177% | 0.188% |
| | ±0% | ±3.717% | ±0% | ±0.001% | ±0% | ±0% | ±0.225% | ±0.017% | ±0.027% |
| Slovenia | 0.823% | 0.014% | 0.114% | 0.016% | 0.105% | 0.019% | 0.090% | 0.035% | 0.036% |
| | ±0.827% | ±0% | ±0.082% | ±0% | ±0% | ±0% | ±0.007% | ±0.003% | ±0.006% |
| Somalia | 9.245% | 4.022% | 0.111% | 0.030% | 0.120% | 0.031% | 0.109% | 0.048% | 0.058% |
| | ±37.350% | ±19.591% | ±0% | ±0% | ±0% | ±0% | ±0.006% | ±0.004% | ±0.034% |
| South Africa | 0.133% | 10.777% | 0.027% | 0.027% | 0.029% | 0.021% | 0.032% | 0.033% | 0.133% |
| | ±0.001% | ±52.681% | ±0% | ±0% | ±0% | ±0.002% | ±0.003% | ±0.014% | ±0.052% |
| South Korea | 0.499% | 1.277% | 0.312% | 0.312% | 0.193% | 0.193% | 0.203% | 0.202% | 0.209% |
| | ±0.033% | ±4.037% | ±0% | ±0% | ±0% | ±0% | ±0.007% | ±0.008% | ±0.013% |
| South Sudan | 0.463% | 0.745% | 0.421% | 0.473% | 0.408% | 0.401% | 0.411% | 0.419% | 0.426% |
| | ±0% | ±1.657% | ±0% | ±0.285% | ±0% | ±0.002% | ±0.002% | ±0.005% | ±0.010% |
| Spain | 0.993% | 4.113% | 0.610% | 0.059% | 0.029% | 0.019% | 0.026% | 0.019% | 0.021% |
| | ±0% | ±19.573% | ±0.480% | ±0% | ±0% | ±0% | ±0.002% | ±0% | ±0.005% |
| Switzerland | 2.926% | 0.098% | 1.141% | 1.955% | 0.019% | 0.014% | 0.018% | 0.015% | 0.016% |
| | ±10.493% | ±0.001% | ±4.370% | ±8.749% | ±0% | ±0% | ±0.001% | ±0% | ±0.002% |
| Thailand | 4.522% | 6.959% | 0.864% | 0.864% | 0.558% | 0.558% | 0.581% | 0.635% | 0.596% |
| | ±19.019% | ±25.233% | ±0% | ±0% | ±0% | ±0% | ±0.030% | ±0.057% | ±0.064% |
| Tunisia | 0.940% | 0.087% | 0.079% | 0.062% | 0.038% | 0.038% | 0.045% | 7.281% | 0.039% |
| | ±2.267% | ±0% | ±0% | ±0% | ±0% | ±0% | ±0.028% | ±35.480% | ±0.001% |
| United Kingdom | 1.042% | 0.133% | 0.491% | 0.234% | 0.014% | 0.011% | 0.020% | 0.011% | 0.022% |
| | ±0% | ±0.030% | ±0% | ±0.803% | ±0% | ±0% | ±0.028% | ±0.001% | ±0.039% |
| Zimbabwe | 0.974% | 0.741% | 0.606% | 0.606% | 0.341% | 0.341% | 5.201% | 1.135% | 0.475% |
| | ±2.984% | ±1.691% | ±0% | ±0% | ±0% | ±0% | ±23.363% | ±3.296% | ±0.086% |
| **Wins** | 0 | 6 | **0** | **10** | 7 | **32** | 0 | **1** | 5 |

5. $\theta \in [0, 0.5]$;

6. $\lambda \in [0.0001, 0.5]$;

7. $\mu \in [0, 8]$.

The idea behind an additional experiment with eSEIUR[range] was to check if the chosen search ranges are sufficient for fitting.

The results in Table 3 show that the modified models were better in most cases; for example, the eSEIR model was the best for 8 countries out of 61. Comparing the SI and Bass diffusion models, the second was the best among all other models for 7 countries out of 61, while the SI model was not the best for country. Between the SIR and eSIR models, the modified model and appears to be the best model for 9 countries. The models with unreported cases did not show best results for any of the countries; however, the difference in relative error was not large. Some of the values in Table 3 are relatively large, especially for SI and BD model, where standard deviation of errors could reach 100%. This is due to the stochastic nature of the

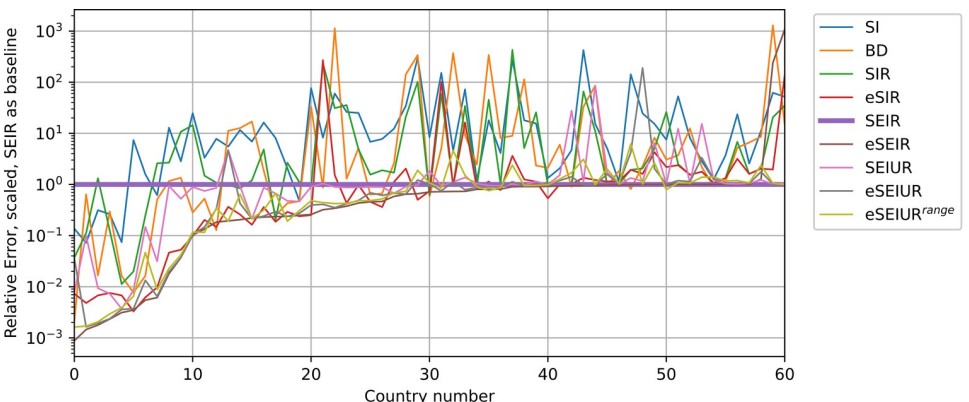

**Fig 15. Scaled relative error values of tested models, ordered by eSEIR.**

L-SHADE-R algorithm, which may not find a good fit in all 25 runs. It can be fixed by restarting the method or running it until it gets a reasonable error value. Large mean and standard deviations are usually due to a single outlier in error values. We leave these values in the Table 3 for the reader to be aware, that there could be some unsuccessful optimization attempts due to randomness of the approach. However, at the end the results of the best fit are considered.

Fig 15 provides the visualization of the relative error values of the basic and modified models of all countries. Here the error values of the SEIR model are chosen as baseline, i.e. the error values of all models are divided by error value of SEIR of each particular country. Also, the countries are sorted by eSEIR relative error values to improve visibility.

Considering the results in Fig 15, it can be concluded that the SI model is not sufficient for describing the dynamics of the infection spread, however, the BD model is much better, and is capable of outperforming other models, including SEIR in many cases. The eSEIR model performs much better than SEIR in up to 40 countries on average, and SEIUR has similar performance to standard SEIR. As for eSEIUR and eSEIUR$^{range}$ models, the former has better performance in many cases, but the difference is not large.

Table 4 contains the aggregated pairwise comparison results of the standard and modified models across all countries. The comparison was performed using non-parametric Mann–Whitney statistical U tests with normal approximation and tie-breaking, as well as a significance level $p = 0.01$. For every country, the comparison was performed using the results of 25 runs, and the following three outcomes were considered: the modified model was better, equal, or worse than the original. The numbers of better, equal, and worse cases were summed together.

The results in Table 4 prove that the modified models were better in most cases. In particular, the BD model was better than SI for 57 countries out of 61. The modified SIR model was worse than original only for one country. The modification also significantly improved the SEIR model, where the original model was statistically equal only for 13 cases out of 61, for all

**Table 4. Statistical comparison of the basic and modified models using Mann–Whitney U tests.**

| Algorithms | BD vs. SI | eSIR vs. SIR | eSEIR vs. SEIR | eSEIUR vs. SEIUR | eSEIUR$^{range}$ vs. eSEIUR |
|---|---|---|---|---|---|
| Better | 57 | 51 | 43 | 32 | 5 |
| Equal | 4 | 9 | 13 | 20 | 39 |
| Worse | 0 | 1 | 0 | 9 | 17 |

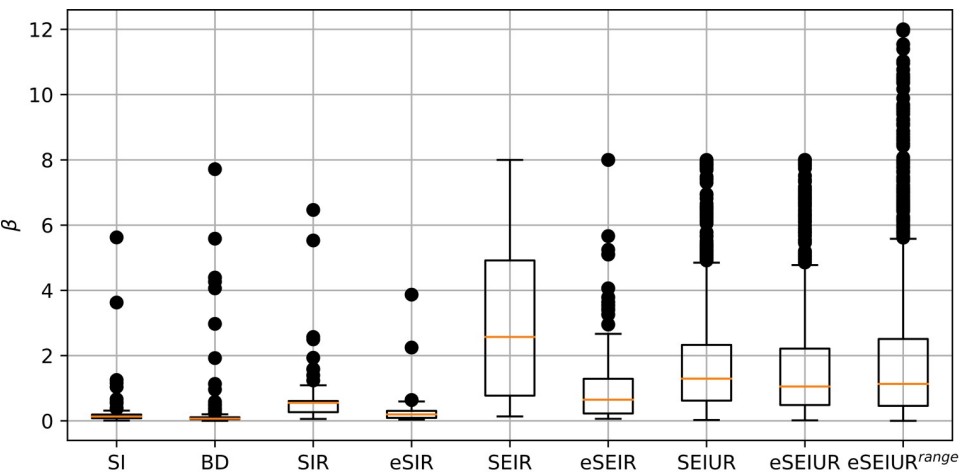

**Fig 16. Values of $\beta$ parameter, results from all runs on all countries.**

other cases the modified model was better. Comparing eSEIUR and eSEIUR$^{range}$, it can be seen that expanding the search range for parameter values results in equal or worse error values, because bigger range results in larger search space, hence lower overall performance of optimization algorithm. This is important for the numerical experiments, which is highly dependent on the parameters' boundaries.

To estimate whether the ranges of the parameters' search are appropriate, the boxplots were drawn for the $\beta$ parameter for all considered models. The $\beta$ parameter was chosen, as it is present in all models, and has significant influence, as it is responsible for new infections caused by contacts between infected and susceptible. Fig 16 shows box plots for all models, the parameter values were taken from all 25 runs on all countries.

As can be seen from Fig 16, for simple models like SI and SIR, large $\beta$ values are not required, so the range from 0 to 8 is appropriate. As for SEIR and eSEIR models, they also stay within this range, and even largest values rarely reach 5. The SEIUR and eSEIUR models, on the other hand, reach the maximum value, and the average value of $\beta$ may be larger than 2. As for the experiment with larger range, it can be seen that although some parameter reach value 8 or even more, most of them stay below 4.

Table 5 contains the calculated basic reproduction rates, $R0$, for every country for the SIR and eSIR models. For the SI and BD models, the calculation of $R0$ was not possible, as the recovery of the infected was not considered in these models. Due to the complexity of the SEIR and SEIUR models, the $R0$ numbers were not estimated for in these cases. The values in Table 5 are calculated based on the found parameter values using the best out of 25 independent runs as $R0 = \beta/\gamma$ for the SIR and eSIR model.

The results in Table 5 show that the calculated $R0$ values for SIR model was 1.647, and for eSIR model– 2.367. For some countries, the values of $R0$ were far from realistic values; however, most of them were within the expected range. According to some of the initial estimations in [29], the $R0$ for COVID-19 was estimated by a meta-analysis as 3.42. It should be noted that the R0 values for the modified model are on average larger than for the original SIR. This is because part of the dynamics in the modified model is described by added exponential term which influences the change of the ratio between β and γ which determine the value of R0. From epidemiological point of view this means that if there is an additional source of infection except contacts between susceptible and infected, intensity of the infection spread would be higher due to the fact, that each susceptible person would have more chances to be infected

**Table 5. Reproduction rates, *R*0, for all countries.**

| Country | SIR | eSIR |
|---|---|---|
| Afghanistan | 1.488 | 3.004 |
| Armenia | 1.172 | 3.339 |
| Australia | 5.545 | 5.545 |
| Austria | 1.759 | 2.561 |
| Azerbaijan | 1.225 | 1.225 |
| Belarus | 1.173 | 1.315 |
| Belgium | 1.343 | 1.174 |
| Bolivia | 1.108 | 1.243 |
| Burkina Faso | 1.119 | 0.983 |
| Cambodia | 0.843 | 0.576 |
| Cameroon | 1.111 | 2.379 |
| Canada | 1.233 | 1.106 |
| Chile | 1.154 | 1.128 |
| Costa Rica | 1.258 | 1.732 |
| Cote d'Ivoire | 1.103 | 2.892 |
| Croatia | 1.268 | 1.695 |
| Cuba | 1.166 | 1.154 |
| Czechia | 1.351 | 1.952 |
| DRC | 1.510 | 2.439 |
| Egypt | 1.501 | 1.496 |
| El Salvador | 1.192 | 3.129 |
| Estonia | 1.288 | 1.185 |
| Finland | 1.214 | 1.747 |
| France | 1.319 | 1.276 |
| Germany | 2.325 | 2.623 |
| Greece | 1.222 | 1.454 |
| Haiti | 2.588 | 2.761 |
| Hungary | 1.211 | 1.712 |
| India | 1.107 | 2.273 |
| Iraq | 1.081 | 1.637 |
| Ireland | 1.297 | 2.802 |
| Israel | 1.361 | 2.114 |
| Italy | 1.278 | 1.166 |
| Japan | 4.739 | 4.739 |
| Latvia | 1.197 | 1.040 |
| Lebanon | 1.727 | 2.047 |
| Liberia | 2.136 | 3.131 |
| Lithuania | 1.207 | 0.975 |
| Mali | 1.142 | 1.142 |
| Mauritania | 1.164 | 1.196 |
| Mauritius | 4.234 | 1.275 |
| Netherlands | 1.344 | 1.178 |
| New Zealand | 1.505 | 4.587 |
| Norway | 1.359 | 1.181 |
| Oman | 1.439 | 3.777 |
| Pakistan | 1.160 | 3.883 |
| Saudi Arabia | 1.144 | 2.462 |

(*Continued*)

**Table 5.** (Continued)

| Country | SIR | eSIR |
|---|---|---|
| Serbia | 1.422 | 2.426 |
| Singapore | 1.104 | 2.285 |
| Slovakia | 1.439 | 5.880 |
| Slovenia | 1.270 | 4.905 |
| Somalia | 1.126 | 2.803 |
| South Africa | 1.268 | 4.112 |
| South Korea | 2.970 | 2.912 |
| South Sudan | 3.547 | 5.771 |
| Spain | 1.367 | 2.290 |
| Switzerland | 1.447 | 1.959 |
| Thailand | 4.590 | 4.590 |
| Tunisia | 1.244 | 1.755 |
| UK | 1.229 | 2.263 |
| Zimbabwe | 3.019 | 3.019 |
| **Average** | 1.647 | 2.367 |

in certain period of time. Although R0 in Table 5 is not always larger for eSIR then for SIR, this is clearly the observed trend.

Fig 17 provides an example of the fitted curves on the real data for Saudi Arabia.

In Fig 17, the basic models, such as SI, SIR, and SEIR are shown to be incapable of capturing the real dynamics, while even the simple BD was enough to achieve an acceptable fit to the real data. The modified models, which incorporated the exponential outflow, were better than their non-modified versions. The results in Fig 17 were obtained with the parameter values, shown in Table 6.

In Table 6 it can be seen that $\theta$ parameter is significantly smaller than the $\beta$ parameter. This is due to the fact that the total population is considered as a normalization constant.

Table 7 contains the start date, end date, initial and final numbers of infected, and population size for every country used in the computations.

Table 7 is provided to assure the repeatability of similar simulation experiments on the same dataset with different models.

The values in Tables 6 and 7, combined with the description of all models can be used to reproduce the performed experiments.

## Discussion

The task was to perform parameterization of classical models to obtain parameters for COVID-19 to improve the knowledge about modelling such systems. At the beginning, aside from the classical models like SI, SIR, SEIR, and SEIUR, the Bass diffusion model was added, which is rarely used in the field of epidemiology [15]. Surprisingly, the Bass diffusion model provided fits as good as the more advanced models, such as SEIR. This led to the idea of applying additional exponential outflow from the susceptible to modify the known models. The main idea behind this additional exponential outflow modification was that the transitions from susceptible to infected or exposed would also occur without direct interactions between the susceptible and infected, with the infection caused by some external factor. With the identification of best models, which incorporated the exponential outflow from the Bass diffusion model, it was possible to improve the forecasting of the first wave of the epidemics. The possible transfer to use in analysing the second, third and other waves is a topic for further research.

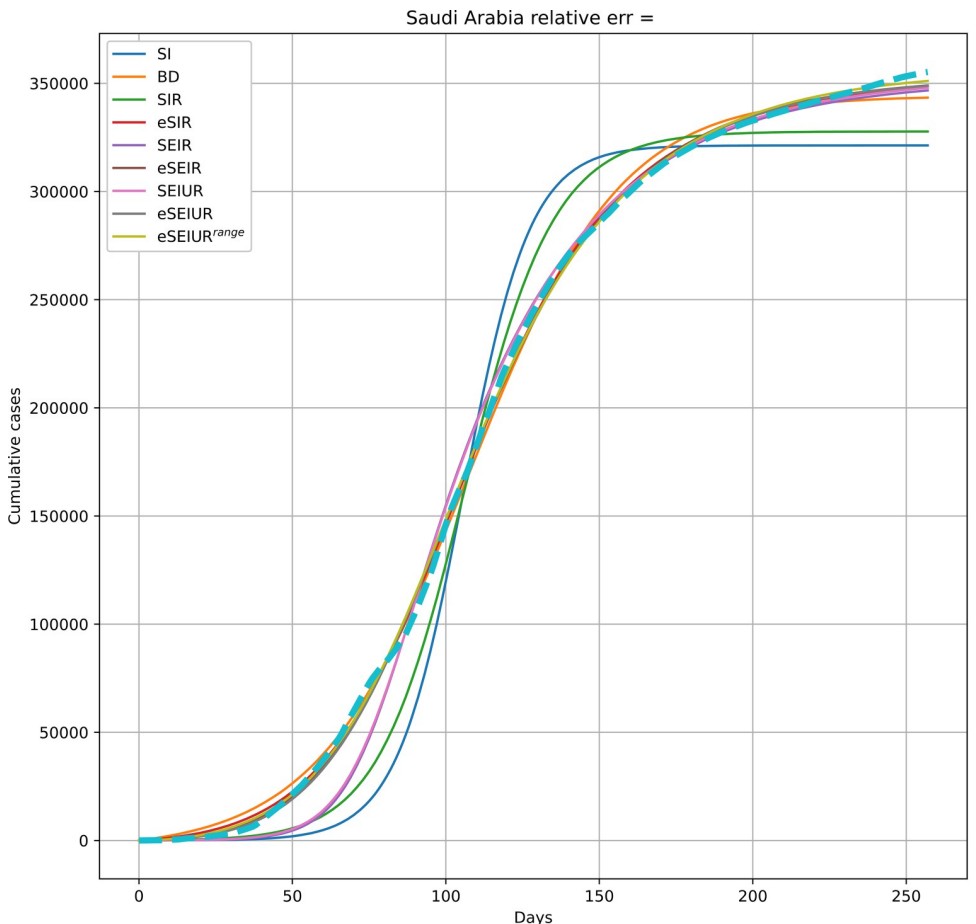

**Fig 17. Example of fitting all models to real data on Saudi Arabia.**

**Table 6. Parameters for all models in Fig 17.**

| SI | $\beta$ | $\alpha$ | | | | | |
|---|---|---|---|---|---|---|---|
| | 0.09164 | 0.00923 | | | | | |
| BD | $\beta$ | $\theta$ | $\alpha$ | | | | |
| | 0.03998 | 0.00051 | 0.00989 | | | | |
| SIR | $\beta$ | $\gamma$ | $\alpha$ | | | | |
| | 0.57224 | 0.50000 | 0.03899 | | | | |
| eSIR | $\beta$ | $\gamma$ | $\theta$ | $\alpha$ | | | |
| | 0.07407 | 0.03008 | 0.00022 | 0.01133 | | | |
| SEIR | $\beta$ | $\sigma$ | $\gamma$ | $\alpha$ | | | |
| | 3.40981 | 0.02381 | 0.50000 | 0.01011 | | | |
| eSEIR | $\beta$ | $\sigma$ | $\gamma$ | $\theta$ | $\alpha$ | | |
| | 1.07283 | 0.04749 | 0.50000 | 0.00041 | 0.01851 | | |
| SEIUR | $\beta$ | $\sigma$ | $\gamma$ | $\alpha$ | $\lambda$ | $\mu$ | |
| | 1.43935 | 0.02387 | 0.48827 | 0.02686 | 0.37805 | 2.93869 | |
| | $\beta$ | $\sigma$ | $\gamma$ | $\theta$ | $\alpha$ | $\lambda$ | $\mu$ |
| eSEIUR | 0.20732 | 0.03195 | 0.13148 | 0.00071 | 0.04160 | 0.25001 | 2.66800 |
| eSEIUR$^{range}$ | 0.30932 | 0.02524 | 0.02165 | 0.00140 | 0.14971 | 0.06830 | 0.62859 |

**Table 7. Start, end date, and number of infected for all countries.**

| Country | Start date | Initial infected | End date | Infected at end | Population |
|---|---|---|---|---|---|
| Afghanistan | 2020-03-13 | 13 | 2020-08-30 | 38155 | 38928341 |
| Armenia | 2020-03-15 | 26 | 2020-08-30 | 43750 | 2963234 |
| Australia | 2020-02-01 | 12 | 2020-04-30 | 6766 | 25499881 |
| Austria | 2020-03-02 | 18 | 2020-05-02 | 15558 | 9006400 |
| Azerbaijan | 2020-03-15 | 23 | 2020-08-10 | 33647 | 10139175 |
| Belarus | 2020-03-14 | 27 | 2020-08-14 | 69308 | 9449321 |
| Belgium | 2020-03-04 | 23 | 2020-06-11 | 59711 | 11589616 |
| Bolivia | 2020-03-16 | 11 | 2020-11-28 | 144592 | 11673029 |
| Burkina Faso | 2020-03-18 | 20 | 2020-06-11 | 892 | 20903278 |
| Cambodia | 2020-03-19 | 37 | 2020-08-09 | 251 | 16718971 |
| Cameroon | 2020-03-19 | 13 | 2020-10-05 | 20924 | 26545864 |
| Canada | 2020-03-02 | 27 | 2020-07-04 | 107185 | 37742157 |
| Chile | 2020-03-03 | 10 | 2020-12-11 | 567974 | 19116209 |
| Costa Rica | 2020-03-11 | 13 | 2021-03-10 | 207832 | 5094114 |
| Cote d'Ivoire | 2020-03-21 | 14 | 2020-11-08 | 20832 | 26378275 |
| Croatia | 2020-03-13 | 32 | 2020-05-15 | 2222 | 4105268 |
| Cuba | 2020-03-21 | 21 | 2020-06-23 | 2318 | 11326616 |
| Czechia | 2020-03-06 | 18 | 2020-05-04 | 7819 | 10708982 |
| DRC | 2020-03-21 | 23 | 2020-09-21 | 10519 | 89561404 |
| Egypt | 2020-03-08 | 49 | 2020-08-22 | 97237 | 102334403 |
| El Salvador | 2020-03-27 | 13 | 2020-10-13 | 30480 | 6486201 |
| Estonia | 2020-03-08 | 10 | 2020-05-30 | 1865 | 1326539 |
| Finland | 2020-03-07 | 15 | 2020-06-21 | 7143 | 5540718 |
| France | 2020-02-28 | 57 | 2020-05-20 | 183130 | 68147687 |
| Germany | 2020-02-03 | 12 | 2020-05-31 | 183410 | 83783945 |
| Greece | 2020-03-07 | 46 | 2020-05-27 | 2903 | 10423056 |
| Haiti | 2020-04-01 | 15 | 2020-11-01 | 9054 | 11402533 |
| Hungary | 2020-03-10 | 9 | 2020-06-18 | 4079 | 9660350 |
| India | 2020-03-06 | 31 | 2021-02-14 | 10916589 | 1380004385 |
| Iraq | 2020-03-02 | 26 | 2021-01-30 | 618922 | 40222503 |
| Ireland | 2020-03-08 | 19 | 2020-06-23 | 25391 | 4937796 |
| Israel | 2020-03-04 | 15 | 2020-05-15 | 16523 | 8655541 |
| Italy | 2020-02-23 | 155 | 2020-06-18 | 238159 | 60461828 |
| Japan | 2020-02-01 | 20 | 2020-06-16 | 17484 | 126476458 |
| Latvia | 2020-03-11 | 10 | 2020-06-12 | 1096 | 1886202 |
| Lebanon | 2020-03-03 | 13 | 2021-03-24 | 448721 | 6825442 |
| Liberia | 2020-04-06 | 14 | 2020-09-04 | 1306 | 5057677 |
| Lithuania | 2020-03-17 | 19 | 2020-06-30 | 1812 | 2722291 |
| Mali | 2020-03-29 | 18 | 2020-08-03 | 2543 | 20250834 |
| Mauritania | 2020-05-15 | 29 | 2020-10-10 | 7550 | 4649660 |
| Mauritius | 2020-03-22 | 28 | 2020-10-30 | 441 | 1271767 |
| Netherlands | 2020-03-02 | 18 | 2020-05-31 | 46645 | 17134873 |
| New Zealand | 2020-03-19 | 28 | 2020-06-09 | 1504 | 4822233 |
| Norway | 2020-03-01 | 19 | 2020-05-15 | 8219 | 5421242 |
| Oman | 2020-03-05 | 16 | 2020-09-09 | 87939 | 5106622 |
| Pakistan | 2020-03-11 | 20 | 2020-09-04 | 298025 | 220892331 |
| Saudi Arabia | 2020-03-10 | 20 | 2020-11-22 | 355258 | 34813867 |

*(Continued)*

**Table 7.** (Continued)

| Country | Start date | Initial infected | End date | Infected at end | Population |
|---|---|---|---|---|---|
| Serbia | 2020-03-13 | 35 | 2020-06-02 | 11454 | 6804596 |
| Singapore | 2020-02-01 | 16 | 2020-10-08 | 57849 | 5850343 |
| Slovakia | 2020-03-12 | 16 | 2020-05-30 | 1521 | 5459643 |
| Slovenia | 2020-03-08 | 16 | 2020-05-22 | 1468 | 2078932 |
| Somalia | 2020-04-12 | 25 | 2020-08-10 | 3227 | 15893219 |
| South Africa | 2020-03-12 | 17 | 2020-09-28 | 671669 | 59308690 |
| South Korea | 2020-02-02 | 15 | 2020-05-04 | 10804 | 51269183 |
| South Sudan | 2020-04-30 | 35 | 2020-07-11 | 2021 | 11193729 |
| Spain | 2020-02-28 | 32 | 2020-05-22 | 234824 | 46754783 |
| Switzerland | 2020-02-29 | 18 | 2020-04-24 | 28677 | 8654618 |
| Thailand | 2020-01-30 | 14 | 2020-06-26 | 3162 | 69799978 |
| Tunisia | 2020-03-15 | 18 | 2020-05-22 | 1048 | 11818618 |
| UK | 2020-02-05 | 9 | 2020-07-01 | 285279 | 67886004 |
| Zimbabwe | 2020-04-17 | 24 | 2020-10-10 | 8010 | 14862927 |

The presented results show that the modified models had better fit to real-world data, although including an additional parameter in the model made it more complex for the optimization method to solve. This means that the models that have additional exponential outflow from susceptible to infected or exposed are more suitable to describe the real dynamics of infections. This would suggest that a significant part of the dynamics is not due to the contact between the susceptible and infected [30], but because of some external factors [31]. One explanation could be that the virus is spreading with aerosol/food or via some unknown mechanism, independent of the contacts between the susceptible and infected. This external source could also be from the unreported immigration of people from infected areas and corresponding accumulation of the virus in aerosol. As in the present study, only the first wave was considered, and it is possible to suggest that even before the pandemic breakout, the number of virus carriers had significantly accumulated, with these being asymptomatic cases, transitioning later with the exponential flow mechanism.

These are only a few hypothetical reasons that could explain this behaviour. In any case, the models imply the presence of such mechanisms, but this should be the topic of further research in the field of epidemiology. If our observations and proposed modifications are confirmed by other independent studies, then the SIR, SEIR, and SEIUR models should be modified as suggested. Moreover, explanations should be provided to outline the related dynamics. The developed models provided guidelines to explain the apparent mechanism, which was not considered in the classical models.

According to [32], "The ability of SARS-CoV-2 to remain viable longer on surfaces taken together with its higher virulence in establishing an infection makes it very likely that this coronavirus uses other modes of transmission in addition to respiratory droplets". This could be one explanation of the observed higher efficiency of the modified models, although some studies, such as [33], suggest that this cannot be the main source of transmission, because the risk of infection after 3 days (72 hours) is minor.

Another explanation could be that the aerosol particles containing respiratory viruses are significantly accumulated in human environments, which are in large extent buildings, and as such this represent independent source of infection, not directly connected to person-to-person transmission, which is indicated in [34–36].

One should be aware, that the spread of infectious disease inevitably involves spatially connected dynamics, which could be modelled in more details. Several studies pointed out that COVID-19 epidemics, being a human-to-human driven disease, evolves locally [37, 38]. The infected persons spread the disease to close susceptible persons, not to any suspected person in the entire country. One should note that this is the limitation of the presented models, which might be used on an abstract, aggregate level.

In this study, the parameterization of the first wave was performed using four classical models and four modified models. The experiments showed that the modification based on the Bass diffusion model allowed for the better fitting of curves, which was also indicated in [15]. By the examination of the equations and the comparison of the modified and unmodified models, one could anticipate that an additional mechanism of COVID-19 spread is present that is not considered in classical models.

## Author Contributions

**Conceptualization:** Vladimir Stanovov, Andrej Škraba.

**Formal analysis:** Andrej Škraba.

**Funding acquisition:** Eugene Semenkin, Andrej Škraba.

**Investigation:** Vladimir Stanovov, Stanko Grabljevec, Shakhnaz Akhmedova.

**Methodology:** Stanko Grabljevec, Shakhnaz Akhmedova, Eugene Semenkin, Andrej Škraba.

**Project administration:** Eugene Semenkin.

**Resources:** Eugene Semenkin, Andrej Škraba.

**Software:** Vladimir Stanovov, Andrej Škraba.

**Supervision:** Stanko Grabljevec, Radovan Stojanović.

**Validation:** Stanko Grabljevec, Radovan Stojanović, Črtomir Rozman.

**Visualization:** Vladimir Stanovov, Andrej Škraba.

**Writing – original draft:** Vladimir Stanovov, Andrej Škraba.

**Writing – review & editing:** Stanko Grabljevec, Radovan Stojanović, Črtomir Rozman.

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
