## [Decision Letter · Decision Letter 0]

29 Jun 2022

PONE-D-22-12244Identification of COVID-19 Spread Mechanisms Based on First-Wave Data, Simulation Models, and Evolutionary AlgorithmsPLOS ONE

Dear Dr. Stanovov,

Thank you for submitting your manuscript to PLOS ONE. After careful consideration, we feel that it has merit but does not fully meet PLOS ONE’s publication criteria as it currently stands. Therefore, we invite you to submit a revised version of the manuscript that addresses the points raised during the review process.

We look forward to receiving your revised manuscript.

Kind regards,

Sebastián Gonçalves, Ph.D.

Academic Editor

PLOS ONE

Journal Requirements:

  "This work was supported by the Ministry of Science and Higher Education of the Russian Federation within limits of state contract № FEFE-2020-0013 and by the Slovenian Research Agency (ARRS) (programs No.: UNI-MB-0586-P5-0018, No.: BI-RU/19-20-034, No. BI-ME/18-20-009)."

 "This work was supported by the Ministry of Science and Higher Education of the Russian Federation within limits of state contract № FEFE-2020-0013 and by the Slovenian Research Agency (ARRS) (programs No.: UNI-MB-0586-P5-0018, No.: BI-RU/19-20-034, No. BI-ME/18-20-009)."

 "This work was supported by the Ministry of Science and Higher Education of the Russian Federation within limits of state contract № FEFE-2020-0013 and by the Slovenian Research Agency (ARRS) (programs No.: UNI-MB-0586-P5-0018, No.: BI-RU/19-20-034, No. BI-ME/18-20-009)."

Reviewers' comments:

Reviewer's Responses to Questions

**Comments to the Author**

1. Is the manuscript technically sound, and do the data support the conclusions?

Reviewer #1: Partly

Reviewer #2: Partly

2. Has the statistical analysis been performed appropriately and rigorously? 

Reviewer #1: No

Reviewer #2: Yes

3. Have the authors made all data underlying the findings in their manuscript fully available?

Reviewer #1: Yes

Reviewer #2: Yes

4. Is the manuscript presented in an intelligible fashion and written in standard English?

Reviewer #1: Yes

Reviewer #2: Yes

5. Review Comments to the Author

Reviewer #1: The article is an important attempt to measure and compare the predictive power of different models in explaining the first wave of COVID-19 in several countries. Extended models that include outflow infections were proposed, resulting in better predictive power than the original (and often used) models. This purpose was very clear, the text is well written, and the models are described in sufficient detail. On the other hand, the fitting process has missing information and arrives at contradictory results. Here are some of the points of statistical and theoretical concern about this issue:

1. It is well known that a system of simultaneous nonlinear differential equations is very sensitive in terms of its initial conditions. However, the initial conditions information is missing.

2. It used a heuristic algorithm to estimate the beginning and the end of the first wave. This algorithm uses some arbitrarily taken parameters. It was not clear if the results are robust by changing these parameters.

3. All extended versions of the well-known SI, SIR, SEIR, and SEUIR models add the new parameter theta. The original SI, SIR, SEIR, and SEUIR models are particular cases of these extended versions when theta goes to zero. So, by construction, the predictive power of the extended versions needs to be at least as good as the original versions (assuming that the same initial condition was used). However, in Table 4, there are some cases when the extended version is worse. This contradiction probably arises from two origins: (i) some limitations of the optimization algorithm itself; (ii) the original and extended versions use different initial conditions. This must be better explained, and a better fitting process must be made.

4. In the fitting process, the parameters are optimized within some boundaries taken as given. It was not explained why these boundaries are chosen and how robust are the results by changing their values.

5. The goodness-of-fit is measured using a normalized version of the Euclidean distance between the actual and the predicted cumulative number of cases. It would be interesting to verify the robustness of the results with other metrics.

6. Finally, several works pointed out that COVID-19 epidemics, being a human-to-human driven disease, evolves locally [1]. The infected persons spread the disease to close susceptible persons, not to any suspected person in the entire country. Hence, the model will not work at the national level. A good evolution of your regression analysis is working at some regional level. Additionally, working at this level, the outflow infections can be easily interpreted as internal migration.

Overall, I consider that the manuscript has some interesting results for the readers of this issue but requires rigorous and careful revision of the fitting process. Conditioned on overcoming the theoretical and statistical problems listed above, I recommend considering a revised version of the manuscript for publication.

[1] Some papers:

Arenas, Alex, et al. "Modeling the spatiotemporal epidemic spreading of COVID-19 and the impact of mobility and social distancing interventions." Physical Review X 10.4 (2020): 041055.

Cardoso, Ben-Hur Francisco, and Sebastián Gonçalves. "Universal scaling law for human-to-human transmission diseases." EPL (Europhysics Letters) 133.5 (2021): 58001.

Reviewer #2: Reference: Manuscript PONE-D-22-12244

` Identification of COVID-19 Spread Mechanisms Based on First-Wave Data, Simulation Models, and Evolutionary Algorithms ‘

In this manuscript, the authors perform a comparative analysis of methods, using traditional and modified epidemiological models, based on differential equations. The model parametrization was done based on evolutionary algorithms in order to find the best fit for the first-wave COVID-19 data in 61 countries around the world. The modified model was inspired in the Bass diffusion model (see references [13] and [15]).

The manuscript is well-written and the application to actual epidemics is well done but there are some points to be clarified as I will discuss below:

1) Concerning the original Bass diffusion model, it is possible to become infected by external sources. Is it reasonable for COVID-19 transmission? Nowadays it is assumed that its transmission is mainly by infected people; the risk of transmission by contaminated surfaces is considered very low. I suggest the authors present some references concerning that discussion, from the epidemiological point of view since it is a central point of the manuscript.

2) Assuming the hypothesis that infection may occurs without contact with infected people, your results show that the modified models, inspired in Bass diffusion model, fit data better than the original ones. My other main question is related to the parametrization: beside the data and the fitting methods, the parameters obey ranges of values related to the dynamics of the disease. May the authors justify the parameters’ ranges of values used in the simulations for the set of considered countries? For instance, $\\beta$ varies from 0 to 4 and $\\theta$ varies from 0 to 0.1, but in the discussion the authors conclude that “significant part of the infection is due to external sources. May the authors discuss this point?

3) In relation to figures 7, 8, 9 and 10, where the usual models are compared to their modified versions, wasn’t it possible to reach the same curve of the modified versions changing the parameter values of the usual models? If it is possible, what are the arguments not to do that procedure? For instance, in Figure 9, changing the value of $\\beta$ of SIR model, the curve of eSIR may be reached.

4) The authors say that they eliminate countries as Denmark, Brazil, Uganda, Vietnam, Poland and Northern Macedonia from their analysis “due to peculiarities in the data”. It is supposed that those peculiarities are different for the cited countries. However, at least in the case of Brazil, it would be relevant to explain its exclusion since it was the second country in the number of reported cases during the first-wave of COVID-19, and for part of the period, it was the first country in the number of deaths due to the disease.

5) Looking at the errors of the results for 61 countries, in table 3, for different models, I think that the authors may include, beside the table, a graphical representation of the errors in order to provide a better visualization for comparison between the countries. Due to the number of countries (61) and the number of models (8), the graphics of errors may be presented in a supplementary material.

6) It was curious that the model with unreported cases do not provide good results. Is it associated to the number of parameters or to its variability. Do the authors have any justification for that result?

7) I suggest the authors include the analytical expressions for the basic reproduction number for all models in order to make the comparison easier for the reader.

8) I recommend that the authors include in caption of figure 6 the fitted values of parameters that lead to the curves exhibited in figure 6. It may help the reproduction of the results

6. PLOS authors have the option to publish the peer review history of their article (what does this mean?). If published, this will include your full peer review and any attached files.

Reviewer #1: No

Reviewer #2: No

---

## [Author Response · Author response to Decision Letter 0]

1 Aug 2022

We would like to thank the reviewers for their insightful suggestions. We have tried our best to address them.

We made the source code, as well as initial data, parameters, errors and generated graphs publicly available in a Github repository: https://github.com/VladimirStanovov/OWID-COVID-19-Analysis

Reviewer #1

1. It is well known that a system of simultaneous nonlinear differential equations is very sensitive in terms of its initial conditions. However, the initial conditions information is missing.

The initial conditions of the models, i.e. state variables, were set according to the available data. First value of the Infected, that were recorded in official statistics, were used for initialization of the Infected state. Number of susceptible was determined according to the population size of each country. In particular, in addition to the mentioned parameters, for every model the algorithm had to tune the ω parameter, responsible for the total number of susceptible at the considered time period. For example, if the total population was 1 million people, setting ω=0.01 meant that the number of susceptible was set to 10000. Other states, such as exposed, recovered and unreported, were initialized to zero.

2. It used a heuristic algorithm to estimate the beginning and the end of the first wave. This algorithm uses some arbitrarily taken parameters. It was not clear if the results are robust by changing these parameters.

Several experiments were conducted to develop and parameterize the proper mechanism to identify first wave. The algorithm was manually checked for all countries. We have added Figure 14 with a short analysis of the influence of τ and wp parameters, controlling the algorithm for determining the first wave.

One should consider that data for some countries were obviously not taken in regular intervals and some irregularities in number of infected cases also occur, like decrease in cumulative number of infected which should not be possible. This was the reason that a general heuristic approach was designed and used for all countries.

The two main parameters controlling the proposed heuristics for determining the first wave are τ and wp. To demonstrate the influence of these values, Figure 14 shows the possible scenarios of determining the first wave with varied parameter values.

3. All extended versions of the well-known SI, SIR, SEIR, and SEUIR models add the new parameter theta. The original SI, SIR, SEIR, and SEUIR models are particular cases of these extended versions when theta goes to zero. So, by construction, the predictive power of the extended versions needs to be at least as good as the original versions (assuming that the same initial condition was used). However, in Table 4, there are some cases when the extended version is worse. This contradiction probably arises from two origins: (i) some limitations of the optimization algorithm itself; (ii) the original and extended versions use different initial conditions. This must be better explained, and a better fitting process must be made.

The optimization algorithm used in the experiments starts with a random set of solutions, and although the search ranges, starting and ending dates are the same, the final results are always different. The search process for standard and modified models is performed in search spaces with different dimensions. Adding more variables results in more local optima.

We have also observed this before the paper submission, however, we considered, that the main information lies in the model structure which has been identified and it’s connected to dynamics and offers certain explanation about the process.

We went further to the fine-tuning of the optimization process. An additional series of experiments has been performed to test if it is possible to improve the results by fine-tuning the standard models. This fine-tuning means that the parameters of the standard model, such as SI, SIR, SEIR or SEIUR are taken as used as a starting point in the population of the DE individuals to search for optimal parameter values of the modified models, the resulting models are marked as BDFT, eSIRFT, eSEIRFT and eSEIURFT. Note that only one of the individuals in DE population was initialized using parameters of standard models, and all other individuals were initialized randomly. The results of these additional experiments have shown that fine-tuning gives either better or at least the same error rates.

4. In the fitting process, the parameters are optimized within some boundaries taken as given. It was not explained why these boundaries are chosen and how robust are the results by changing their values.

The ranges for σ and γ were chosen according to [26], where transmission from exposed to infected and from infected to recovered were determined between 3 and 14 days. For β the search range was set from 0 to 4 according to preliminary experiments and literature, for example in [27] the β parameter was set to from 1.5 to 3.

To estimate whether the ranges of the parameters' search are appropriate, the boxplots were drawn for the β parameter for all considered models. The β parameter was chosen, as it is present in all models, and has significant influence, as it is responsible for new infections caused by contacts between infected and susceptible. Figure 16 shows box plots for all models, the parameter values were taken from all 10 runs on all countries.

 Chu, J. A statistical analysis of the novel coronavirus (COVID-19) in Italy and Spain. PLoS ONE, 2021, 16, https://doi.org/10.1371/journal.pone.0249037

 Schmitt, F.G. An algorithm for the direct estimation of the parameters of the SIR epidemic model from the I(t) dynamics. European Physical Journal plus, 2022, 137, https://doi.org/10.1140/epjp/s13360-021-02237-7

5. The goodness-of-fit is measured using a normalized version of the Euclidean distance between the actual and the predicted cumulative number of cases. It would be interesting to verify the robustness of the results with other metrics.

Preliminary, we carefully examined the fitting process by using several standard measures for fitness function such as Mean Absolute Percent Error (MAPE), Mean Absolute Error (MAE), Mean Squared Error (MSE). However, the proposed relative error measure REc was more stable in case of extreme values e.g. when the number of infected is close to 0.

6. Finally, several works pointed out that COVID-19 epidemics, being a human-to-human driven disease, evolves locally [1]. The infected persons spread the disease to close susceptible persons, not to any suspected person in the entire country. Hence, the model will not work at the national level. A good evolution of your regression analysis is working at some regional level. Additionally, working at this level, the outflow infections can be easily interpreted as internal migration.

[1] Some papers:

Arenas, Alex, et al. "Modeling the spatiotemporal epidemic spreading of COVID-19 and the impact of mobility and social distancing interventions." Physical Review X 10.4 (2020): 041055.

Cardoso, Ben-Hur Francisco, and Sebastián Gonçalves. "Universal scaling law for human-to-human transmission diseases." EPL (Europhysics Letters) 133.5 (2021): 58001.

Abstract models, that were used, consider the possibility of spreading the disease to arbitrary person in the population which is certain limitation of used models. In reality, the infected persons spread the disease to close susceptible persons.

Therefore, one should consider, that human-to-human driven diseases, evolves locally. For better fitting results spatiotemporal spreading of infections should be considered, however, with the available limited dataset, the development of more detailed model could not be performed. This is also the reason that most predictions were based on abstract models such as the ones used in our study.

Development of spatiotemporal models should be addressed in the future with the usage of new technologies, that enable the tracking and monitoring of health data of everyone. This on the other hand raises the question of privacy just to mention one.

The main contribution of the paper lies in the identification of possible new mechanisms that underlies the Covid-19 spread by examining different model structures.

In the discussion section, the following part was added: “One should be aware, that the spread of infectious disease inevitably involves spatially connected dynamics, which could be modelled in more details. Several studies pointed out that COVID-19 epidemics, being a human-to-human driven disease, evolves locally [29, 30]. The infected persons spread the disease to close susceptible persons, not to any suspected person in the entire country. One should note, that this is the limitation of the presented models, which might be used on an abstract, aggregate level.”

 

Reviewer #2

1) Concerning the original Bass diffusion model, it is possible to become infected by external sources. Is it reasonable for COVID-19 transmission? Nowadays it is assumed that its transmission is mainly by infected people; the risk of transmission by contaminated surfaces is considered very low. I suggest the authors present some references concerning that discussion, from the epidemiological point of view since it is a central point of the manuscript.

We added references to the following part of the discussion: “This would suggest that a significant part of the dynamics is not due to the contact between the susceptible and infected [31], but because of some external factors [32].“

The following part has been added in the discussion: “According to [33], “The ability of SARS-CoV-2 to remain viable longer on surfaces taken together with its higher virulence in establishing an infection makes it very likely that this coronavirus uses other modes of transmission in addition to respiratory droplets”. This could be one explanation of the observed higher efficiency of the modified models.“

 Jayaweera, M., Perera, H.R., Gunawardana, B., & Manatunge, J. (2020). Transmission of COVID-19 virus by droplets and aerosols: A critical review on the unresolved dichotomy. Environmental Research, 188, 109819 – 109819, https://doi.org/10.1016/j.envres.2020.109819

 Lewis, D. (2021). COVID-19 rarely spreads through surfaces. So why are we still deep cleaning? Nature, 590 7844, 26-28, https://doi.org/10.1038/d41586-021-00251-4

 Galbadage, T., Peterson, B.M., & Gunasekera, R.S. (2020). Does COVID-19 Spread Through Droplets Alone? Frontiers in Public Health, 8, https://doi.org/10.3389/fpubh.2020.00163

2) Assuming the hypothesis that infection may occurs without contact with infected people, your results show that the modified models, inspired in Bass diffusion model, fit data better than the original ones. My other main question is related to the parametrization: beside the data and the fitting methods, the parameters obey ranges of values related to the dynamics of the disease. May the authors justify the parameters’ ranges of values used in the simulations for the set of considered countries? For instance, $\\beta$ varies from 0 to 4 and $\\theta$ varies from 0 to 0.1, but in the discussion the authors conclude that “significant part of the infection is due to external sources. May the authors discuss this point?

The ranges for σ and γ were chosen according to preliminary experiments and literature [26], where transmission from exposed to infected and from infected to recovered were determined between 3 and 14 days. For β the search range was set from 0 to 4 according to preliminary experiments and literature, for example in [27] the β parameter was set to from 1.5 to 3.

To estimate whether the ranges of the parameters search are appropriate, the boxplots were drawn for the β parameter for all considered models. The β parameter was chosen, as it is present in all models, and has significant influence, as it is responsible for new infections caused by contacts between infected and susceptible. Figure 16 shows box plots for all models, the parameter values were taken from all 10 runs on all countries.

 Chu, J. A statistical analysis of the novel coronavirus (COVID-19) in Italy and Spain. PLoS ONE, 2021, 16, https://doi.org/10.1371/journal.pone.0249037

 Schmitt, F.G. An algorithm for the direct estimation of the parameters of the SIR epidemic model from the I(t) dynamics. European Physical Journal plus, 2022, 137, https://doi.org/10.1140/epjp/s13360-021-02237-7

3) In relation to figures 7, 8, 9 and 10, where the usual models are compared to their modified versions, wasn’t it possible to reach the same curve of the modified versions changing the parameter values of the usual models? If it is possible, what are the arguments not to do that procedure? For instance, in Figure 9, changing the value of $\\beta$ of SIR model, the curve of eSIR may be reached.

For particular cases, we could tune the parameters of the standard model to closely fit to the modified model, but it is not possible to get equal response, especially for more complex models.

This could also be examined by analytical solutions which, for the simple cases, might be obtained. The analytical solutions are different thus it is not possible to get exactly the same response, however close fit could be possible.

4) The authors say that they eliminate countries as Denmark, Brazil, Uganda, Vietnam, Poland and Northern Macedonia from their analysis “due to peculiarities in the data”. It is supposed that those peculiarities are different for the cited countries. However, at least in the case of Brazil, it would be relevant to explain its exclusion since it was the second country in the number of reported cases during the first-wave of COVID-19, and for part of the period, it was the first country in the number of deaths due to the disease.

Brazil data did not exercise the clear border between first and second wave same as some other listed countries. As long as our goal was to work with the first wave it was not suitable for the analysis. The data of mentioned countries is still valid though not suitable to examine the dynamics of the first wave, which should be clearly present in data.

5) Looking at the errors of the results for 61 countries, in table 3, for different models, I think that the authors may include, beside the table, a graphical representation of the errors in order to provide a better visualization for comparison between the countries. Due to the number of countries (61) and the number of models (8), the graphics of errors may be presented in a supplementary material.

We have added Figure 15, where the SEIR model is chosen as baseline, and other models’ relative errors are scaled. Also, the countries are ordered for better visibility.

6) It was curious that the model with unreported cases do not provide good results. Is it associated to the number of parameters or to its variability. Do the authors have any justification for that result?

In order to compensate the complexity of optimization problems in cases like SEIUR model, the number of generations of DE was increased to 600. According to Figure 15, the models with unreported cases perform similarly to the SEIR and eSEIR models, although a little larger error values were observed. Our justification is that there are several local optima in the search space, and the optimization process converges to one of them. These local optima generate similar dynamics, each providing close fit with the data. The complexity of models with unreported cases adds to the variability, and significantly increases the number of possible local optima, whereas for models without it the optimization converges faster.

7) I suggest the authors include the analytical expressions for the basic reproduction number for all models in order to make the comparison easier for the reader.

We have provided the reproduction number R0=β/γ only for the SIR model based on the estimated parameters. Determination of R0 number for SEIR and SEIUR models is challenging according to several studies (Heng et al., 2020) and it was not in the main focus of our study. For SEIR and SEIUR models there are analytical solutions, however, they require the death rate to be considered, which was not the case in our study. If the death rate is set to 0, these equations simplify to R0=β/γ, however, as β and γ are in different equations, the results of such calculation are often unstable, i.e. R0 could be as large as 30, or as small as 0.3.

Heng, K., & Althaus, C.L. (2020). The approximately universal shapes of epidemic curves in the Susceptible-Exposed-Infectious-Recovered (SEIR) model. Scientific Reports, 10.

8) I recommend that the authors include in caption of figure 6 the fitted values of parameters that lead to the curves exhibited in figure 6. It may help the reproduction of the results

We have added Table 7 with all parameter values to reproduce the graphs in Figure 17 (the figure numbers were changed). Added: “The values in Tables 7 and 8, combined with the description of all models can be used to reproduce the performed experiments.”

---

## [Decision Letter · Decision Letter 1]

31 Aug 2022

PONE-D-22-12244R1Identification of COVID-19 Spread Mechanisms Based on First-Wave Data, Simulation Models, and Evolutionary AlgorithmsPLOS ONE

Dear Dr. Stanovov,

Thank you for submitting your manuscript to PLOS ONE. After careful consideration, we feel that it has merit but does not fully meet PLOS ONE’s publication criteria as it currently stands. Therefore, we invite you to submit a revised version of the manuscript that addresses the points raised during the review process.

 Notice that the issues questioned by the reviewers in this iteration were already raised in the first reports but not properly addressed yet.

We look forward to receiving your revised manuscript.

Kind regards,

Sebastián Gonçalves, Ph.D.

Academic Editor

PLOS ONE

Journal Requirements:

Reviewers' comments:

Reviewer's Responses to Questions

**Comments to the Author**

1. If the authors have adequately addressed your comments raised in a previous round of review and you feel that this manuscript is now acceptable for publication, you may indicate that here to bypass the “Comments to the Author” section, enter your conflict of interest statement in the “Confidential to Editor” section, and submit your "Accept" recommendation.

Reviewer #1: (No Response)

Reviewer #2: (No Response)

2. Is the manuscript technically sound, and do the data support the conclusions?

Reviewer #1: Partly

Reviewer #2: Yes

3. Has the statistical analysis been performed appropriately and rigorously? 

Reviewer #1: No

Reviewer #2: Yes

4. Have the authors made all data underlying the findings in their manuscript fully available?

Reviewer #1: Yes

Reviewer #2: Yes

5. Is the manuscript presented in an intelligible fashion and written in standard English?

Reviewer #1: Yes

Reviewer #2: Yes

6. Review Comments to the Author

Reviewer #1: The article is a significant attempt to measure and compare the predictive power of different models in explaining the first wave of COVID-19 in several countries. Extended models that include outflow infections were proposed, resulting in better predictive power than the original (and often used) models.

The authors answered, in sufficient detail, the six questions I did in my previous report in "Response to Reviewers." However, the authors did not appropriately address, in the Manuscript, the most crucial one: "[...] by construction, the predictive power of the extended versions needs to be at least as good as the original versions[...]"

In "Response to Reviewers," the authors said that they had already recognized this problem, and, in the Manuscript, they included new "fine-tuned" regressions that partially solve this problem. However, all other figures and tables still have this problem. That is not a minor problem. The extended versions need, by construction, to be at least as good as the original versions. By construction, the convergence criterium of regression methods that do not satisfy this condition is incorrect. Having results showing the opposite calls into question the validity of the regression method and, therefore, all their findings.

Overall, the Manuscript has some exciting results for the readers but requires careful revision of this regression process. Conditioned to overcome this statistical problem, I recommend considering a revised version of the Manuscript for publication.

Reviewer #2: Reference: Manuscript PONE-D-22-12244

` Identification of COVID-19 Spread Mechanisms Based on First-Wave Data, Simulation Models, and Evolutionary Algorithms ‘

In the revised version of the manuscript, the authors answer mostly the raised points as well as improved it including some new graphics as the relative error ordered by SEIR model (Figure 15), the box plot of the errors (Figure 16) as well as the table with parameter values assumed for each model at least for an example (Table 7), and the table of errors resulted for new numerical experiments (Table 4). However, some answers are not enough to clarify the reader in my point of view, and the introduced changes lead to some new comments; so I suggest below to complete and clarify them before the manuscript may be published in PLOS One Journal.

1) Since the infection by external sources is responsible of introducing, in the modified models, a term of the progression from susceptible to infected compartment without interaction between them, I think that the authors have not called the attention to the low risk associated to surface transmission. According to more recent report of Centre for Disease Control and Prevention (CDC) (see https://www.cdc.gov/coronavirus/2019-ncov/more/science-and-research/surface-transmission.html), although “people can be infected with SARS-CoV-2 through contact with surfaces”, “surface transmission is no the main root of it”. I suggest they include more recent references calling attention to that observation mainly because it justifies the difference between the assumed values, for instance, in Table 7, between $\\beta$ and $\\theta$ values for all considered models ($\\beta$ is 100 to 1.000 times larger than $\\theta$). I suggest the authors highlight that difference relating the epidemiological feature to the extension of the model.

2) Concerning the range of parameter values of $\\beta$, in reference [27], they considered the epidemic curves of Italy and Spain assuming $beta$ varies from 1.5 to 3; in this manuscript there are epidemic curves of 61 countries the authors have extended the range from 0 to 4. With the aim of checking the fine tuning, in the revised version the authors have performed a set of experiments varying $\\beta$ from 0 to 10 as well as the range of other parameters, leading to new results exposed in Table 4. Differently from other parameters, in my point of view, the parameters $\\beta$ and $\\theta$ (it is possible to assume a relation between them according to comment 1 above) may vary significantly; since the values are not exposed for all countries, it is not possible to check. It would be nice to know, at least, what is the maximal fitted value of $\\beta$ in order to figure out how significant is the extension of $\\beta$ range of values and if the upper limit was enough to describe all countries.

3) The authors pointed out that the optimization problems may lead to unusual situations such as the model with unreported compartment may result to larger errors than the model without it, as pointed out by me and, the extended versions of the models - with progression from susceptible to infected compartments without interaction between them – produces worse results than the usual ones as pointed out by the other reviewer. Since the technical reasons leads to counter-intuitive results, I think it is important to include a sentence about it, maybe in the results section or in the discussion section.

7. PLOS authors have the option to publish the peer review history of their article (what does this mean?). If published, this will include your full peer review and any attached files.

Reviewer #1: No

Reviewer #2: No

---

## [Author Response · Author response to Decision Letter 1]

11 Oct 2022

We would like to thank the reviewers for their insightful suggestions. We have tried our best to address them.

Reviewer #1: The article is a significant attempt to measure and compare the predictive power of different models in explaining the first wave of COVID-19 in several countries. Extended models that include outflow infections were proposed, resulting in better predictive power than the original (and often used) models.

The authors answered, in sufficient detail, the six questions I did in my previous report in "Response to Reviewers." However, the authors did not appropriately address, in the Manuscript, the most crucial one: "[...] by construction, the predictive power of the extended versions needs to be at least as good as the original versions[...]"

In "Response to Reviewers," the authors said that they had already recognized this problem, and, in the Manuscript, they included new "fine-tuned" regressions that partially solve this problem. However, all other figures and tables still have this problem. That is not a minor problem. The extended versions need, by construction, to be at least as good as the original versions. By construction, the convergence criterium of regression methods that do not satisfy this condition is incorrect. Having results showing the opposite calls into question the validity of the regression method and, therefore, all their findings.

Overall, the Manuscript has some exciting results for the readers but requires careful revision of this regression process. Conditioned to overcome this statistical problem, I recommend considering a revised version of the Manuscript for publication.

Answer: Thank you for your valuable comments, we redesigned the experimental setup and used a more advanced optimization method, L-SHADE-R. We have changed the number of runs, which improves confidence of statistical results, and performed experiments with extended search ranges. Thanks to both reviewers’ suggestions, the experimental setup lead to better results, that are logically consisted, i.e. more model parameters enables better fit. In the new version this was experimentally confirmed (see Table 4 and Figure 15). Therefore, we believe that now there are no counterintuitive results in improved version of the paper.

The fine-tuning phase was omitted in the new experimental procedure, because the new optimization method provided better results, hence there was no point in applying fine-tuning.

Also, an error with parameter representation in Table 6 was fixed, where omega parameter was used instead of alpha.

The updated code and results are available at https://github.com/VladimirStanovov/OWID-COVID-19-Analysis

 

Reviewer #2: Reference: Manuscript PONE-D-22-12244

` Identification of COVID-19 Spread Mechanisms Based on First-Wave Data, Simulation Models, and Evolutionary Algorithms ‘

In the revised version of the manuscript, the authors answer mostly the raised points as well as improved it including some new graphics as the relative error ordered by SEIR model (Figure 15), the box plot of the errors (Figure 16) as well as the table with parameter values assumed for each model at least for an example (Table 7), and the table of errors resulted for new numerical experiments (Table 4). However, some answers are not enough to clarify the reader in my point of view, and the introduced changes lead to some new comments; so I suggest below to complete and clarify them before the manuscript may be published in PLOS One Journal.

Answer: Thank you for your valuable comments, we tried our best to address all of them.

1) Since the infection by external sources is responsible of introducing, in the modified models, a term of the progression from susceptible to infected compartment without interaction between them, I think that the authors have not called the attention to the low risk associated to surface transmission. According to more recent report of Centre for Disease Control and Prevention (CDC) (see https://www.cdc.gov/coronavirus/2019-ncov/more/science-and-research/surface-transmission.html), although “people can be infected with SARS-CoV-2 through contact with surfaces”, “surface transmission is no the main root of it”. I suggest they include more recent references calling attention to that observation mainly because it justifies the difference between the assumed values, for instance, in Table 7, between $\\beta$ and $\\theta$ values for all considered models ($\\beta$ is 100 to 1.000 times larger than $\\theta$). I suggest the authors highlight that difference relating the epidemiological feature to the extension of the model.

Answer: We have added several sentences and references about this in the discussion section: “This could be one explanation of the observed higher efficiency of the modified models, although some studies, such as [33], suggest that this cannot be the main source of transmission, because the risk of infection after 3 days (72 hours) is minor. 

Another explanation could be that the aerosol particles containing respiratory viruses are significantly accumulated in human environments, which are in large extent buildings, and as such this represent independent source of infection, not directly connected to person-to-person transmission, which is indicated in [34, 35, 36].”

“In Table 6 it can be seen that θ parameter is significantly smaller than the β parameter. This is due to the fact that the total population is considered as normalization constant.” For example, in SEIR model dS/dt=-βI S/N-θS, β is divided here by N, i.e. normalized by the total population, which is large number, therefore the ratio between β and θ is high.

2) Concerning the range of parameter values of $\\beta$, in reference [27], they considered the epidemic curves of Italy and Spain assuming $beta$ varies from 1.5 to 3; in this manuscript there are epidemic curves of 61 countries the authors have extended the range from 0 to 4. With the aim of checking the fine tuning, in the revised version the authors have performed a set of experiments varying $\\beta$ from 0 to 10 as well as the range of other parameters, leading to new results exposed in Table 4. Differently from other parameters, in my point of view, the parameters $\\beta$ and $\\theta$ (it is possible to assume a relation between them according to comment 1 above) may vary significantly; since the values are not exposed for all countries, it is not possible to check. It would be nice to know, at least, what is the maximal fitted value of $\\beta$ in order to figure out how significant is the extension of $\\beta$ range of values and if the upper limit was enough to describe all countries.

Answer: The ranges of values of β parameter are shown in box plots with average, maximum and minimum values, as well as outliers in Figure 16. There was an error in naming Figure 16, which was previously said to show relative error values, actually it shows β values. We have expanded the ranged for β further to [0, 8], and the experiments have shown that most of the models use smaller values.

3) The authors pointed out that the optimization problems may lead to unusual situations such as the model with unreported compartment may result to larger errors than the model without it, as pointed out by me and, the extended versions of the models - with progression from susceptible to infected compartments without interaction between them – produces worse results than the usual ones as pointed out by the other reviewer. Since the technical reasons leads to counter-intuitive results, I think it is important to include a sentence about it, maybe in the results section or in the discussion section.

Answer: Thank you for your valuable comments, which were similar to other reviewer. We redesigned the experimental setup and used a more advanced optimization method, L-SHADE-R. We have changed the number of runs, which improves confidence of statistical results, and performed experiments with extended search ranges. Thanks to both reviewers’ suggestions, the experimental setup lead to better results, that are logically consisted, i.e. more model parameters enables better fit. In the new version this was experimentally confirmed (see Table 4 and Figure 15). Therefore, we believe that now there are no counterintuitive results in improved version of the paper.

The fine-tuning phase was omitted in the new experimental procedure, because the new optimization method provided better results, hence there was no point in applying fine-tuning.

Also, an error with parameter representation in Table 6 was fixed, where omega parameter was used instead of alpha.

The updated code and results are available at https://github.com/VladimirStanovov/OWID-COVID-19-Analysis

---

## [Decision Letter · Decision Letter 2]

6 Nov 2022

PONE-D-22-12244R2Identification of COVID-19 Spread Mechanisms Based on First-Wave Data, Simulation Models, and Evolutionary AlgorithmsPLOS ONE

Dear Dr. Stanovov,

Thank you for submitting your manuscript to PLOS ONE. After careful consideration, we feel that it has merit but does not fully meet PLOS ONE’s publication criteria as it currently stands. Therefore, we invite you to submit a revised version of the manuscript that addresses the points raised during the review process. Specifically, please address the two points raised by Reviewer #2.

We look forward to receiving your revised manuscript.

Kind regards,

Sebastián Gonçalves, Ph.D.

Academic Editor

PLOS ONE

Journal Requirements:

Reviewers' comments:

Reviewer's Responses to Questions

**Comments to the Author**

1. If the authors have adequately addressed your comments raised in a previous round of review and you feel that this manuscript is now acceptable for publication, you may indicate that here to bypass the “Comments to the Author” section, enter your conflict of interest statement in the “Confidential to Editor” section, and submit your "Accept" recommendation.

Reviewer #1: All comments have been addressed

Reviewer #2: All comments have been addressed

2. Is the manuscript technically sound, and do the data support the conclusions?

Reviewer #1: Yes

Reviewer #2: Yes

3. Has the statistical analysis been performed appropriately and rigorously? 

Reviewer #1: Yes

Reviewer #2: Yes

4. Have the authors made all data underlying the findings in their manuscript fully available?

Reviewer #1: Yes

Reviewer #2: Yes

5. Is the manuscript presented in an intelligible fashion and written in standard English?

Reviewer #1: Yes

Reviewer #2: Yes

6. Review Comments to the Author

Reviewer #1: The article is an important attempt to measure and compare the predictive power of different models in explaining the first wave of COVID-19 in several countries. Extended models that include outflow infections were proposed, resulting in better predictive power than the original (and often used) models. This purpose was very clear, the text is well written, and the models are described in sufficient detail.

The authors answered, in sufficient detail, the questions I did in my previous report.

Overall, the Manuscript has exciting results, and I recommend the Manuscript for publication.

Reviewer #2: Reference: Manuscript PONE-D-22-12244-R2

` Identification of COVID-19 Spread Mechanisms Based on First-Wave Data, Simulation Models, and Evolutionary Algorithms ‘

In the second revised version of the manuscript, the quality of results is improved because the authors used a more advanced optimization method, called -SHADE-R, which produce more consistent results concerning the comparison between models. In my point of view, this new version of manuscript may be published in PLOS One Journal after considering two points:

A) I have observed that, for many countries, there is a significative increasing of R_0 of value from SIR model to eSIR model. Therefore I suggest the authors make a comment about it, from the epidemiological point of view, as well as, if possible, tell us if this pattern is also observed for the other usual and modified models.

B) As far as I understood, the sentence after figure 15 and before table 4 have to be corrected: due to the last column of table 4, eSEIUR model has better perform than eSEIURrange model, so it is necessary to change the sentence “… eSEIUR and eSEIURrange models, the FORMER has better performance in many cases, but the difference is not large. “

7. PLOS authors have the option to publish the peer review history of their article (what does this mean?). If published, this will include your full peer review and any attached files.

Reviewer #1: No

Reviewer #2: No

---

## [Author Response · Author response to Decision Letter 2]

11 Nov 2022

Third review response

We would like to thank the reviewers for their insightful suggestions. We have tried our best to address them.

Reviewer #1: The article is an important attempt to measure and compare the predictive power of different models in explaining the first wave of COVID-19 in several countries. Extended models that include outflow infections were proposed, resulting in better predictive power than the original (and often used) models. This purpose was very clear, the text is well written, and the models are described in sufficient detail.

The authors answered, in sufficient detail, the questions I did in my previous report.

Overall, the Manuscript has exciting results, and I recommend the Manuscript for publication.

Answer: Thank you for the good evaluation of our work.

Reviewer #2: Reference: Manuscript PONE-D-22-12244-R2

` Identification of COVID-19 Spread Mechanisms Based on First-Wave Data, Simulation Models, and Evolutionary Algorithms ‘

In the second revised version of the manuscript, the quality of results is improved because the authors used a more advanced optimization method, called -SHADE-R, which produce more consistent results concerning the comparison between models. In my point of view, this new version of manuscript may be published in PLOS One Journal after considering two points:

A) I have observed that, for many countries, there is a significative increasing of R_0 of value from SIR model to eSIR model. Therefore I suggest the authors make a comment about it, from the epidemiological point of view, as well as, if possible, tell us if this pattern is also observed for the other usual and modified models.

Answer: We have added the following explanation after Table 5: “It should be noted that the R0 values for the modified model are on average larger than for the original SIR. This is because part of the dynamics in the modified model is described by added exponential term which influences the change of the ratio between β and γ which determine the value of R0. From epidemiological point of view this means that if there is an additional source of infection except contacts between susceptible and infected, intensity of the infection spread would be higher due to the fact, that each susceptible person would have more chances to be infected in certain period of time. Although R0 in Table 5 is not always larger for eSIR then for SIR, this is clearly the observed trend.”

In order to answer in full we might add that similar pattern might be observed in other usual and modified models, but in order to support such claim this should thoroughly be checked and also depends on the data; this might be a direction for future research.

B) As far as I understood, the sentence after figure 15 and before table 4 have to be corrected: due to the last column of table 4, eSEIUR model has better perform than eSEIURrange model, so it is necessary to change the sentence “… eSEIUR and eSEIURrange models, the FORMER has better performance in many cases, but the difference is not large. “

Answer: We have corrected this sentence and added “former”.

 

Second review response

We would like to thank the reviewers for their insightful suggestions. We have tried our best to address them.

Reviewer #1: The article is a significant attempt to measure and compare the predictive power of different models in explaining the first wave of COVID-19 in several countries. Extended models that include outflow infections were proposed, resulting in better predictive power than the original (and often used) models.

The authors answered, in sufficient detail, the six questions I did in my previous report in "Response to Reviewers." However, the authors did not appropriately address, in the Manuscript, the most crucial one: "[...] by construction, the predictive power of the extended versions needs to be at least as good as the original versions[...]"

In "Response to Reviewers," the authors said that they had already recognized this problem, and, in the Manuscript, they included new "fine-tuned" regressions that partially solve this problem. However, all other figures and tables still have this problem. That is not a minor problem. The extended versions need, by construction, to be at least as good as the original versions. By construction, the convergence criterium of regression methods that do not satisfy this condition is incorrect. Having results showing the opposite calls into question the validity of the regression method and, therefore, all their findings.

Overall, the Manuscript has some exciting results for the readers but requires careful revision of this regression process. Conditioned to overcome this statistical problem, I recommend considering a revised version of the Manuscript for publication.

Answer: Thank you for your valuable comments, we redesigned the experimental setup and used a more advanced optimization method, L-SHADE-R. We have changed the number of runs, which improves confidence of statistical results, and performed experiments with extended search ranges. Thanks to both reviewers’ suggestions, the experimental setup lead to better results, that are logically consisted, i.e. more model parameters enables better fit. In the new version this was experimentally confirmed (see Table 4 and Figure 15). Therefore, we believe that now there are no counterintuitive results in improved version of the paper.

The fine-tuning phase was omitted in the new experimental procedure, because the new optimization method provided better results, hence there was no point in applying fine-tuning.

Also, an error with parameter representation in Table 6 was fixed, where omega parameter was used instead of alpha.

The updated code and results are available at https://github.com/VladimirStanovov/OWID-COVID-19-Analysis

 

Reviewer #2: Reference: Manuscript PONE-D-22-12244

` Identification of COVID-19 Spread Mechanisms Based on First-Wave Data, Simulation Models, and Evolutionary Algorithms ‘

In the revised version of the manuscript, the authors answer mostly the raised points as well as improved it including some new graphics as the relative error ordered by SEIR model (Figure 15), the box plot of the errors (Figure 16) as well as the table with parameter values assumed for each model at least for an example (Table 7), and the table of errors resulted for new numerical experiments (Table 4). However, some answers are not enough to clarify the reader in my point of view, and the introduced changes lead to some new comments; so I suggest below to complete and clarify them before the manuscript may be published in PLOS One Journal.

Answer: Thank you for your valuable comments, we tried our best to address all of them.

1) Since the infection by external sources is responsible of introducing, in the modified models, a term of the progression from susceptible to infected compartment without interaction between them, I think that the authors have not called the attention to the low risk associated to surface transmission. According to more recent report of Centre for Disease Control and Prevention (CDC) (see https://www.cdc.gov/coronavirus/2019-ncov/more/science-and-research/surface-transmission.html), although “people can be infected with SARS-CoV-2 through contact with surfaces”, “surface transmission is no the main root of it”. I suggest they include more recent references calling attention to that observation mainly because it justifies the difference between the assumed values, for instance, in Table 7, between $\\beta$ and $\\theta$ values for all considered models ($\\beta$ is 100 to 1.000 times larger than $\\theta$). I suggest the authors highlight that difference relating the epidemiological feature to the extension of the model.

Answer: We have added several sentences and references about this in the discussion section: “This could be one explanation of the observed higher efficiency of the modified models, although some studies, such as [33], suggest that this cannot be the main source of transmission, because the risk of infection after 3 days (72 hours) is minor. 

Another explanation could be that the aerosol particles containing respiratory viruses are significantly accumulated in human environments, which are in large extent buildings, and as such this represent independent source of infection, not directly connected to person-to-person transmission, which is indicated in [34, 35, 36].”

“In Table 6 it can be seen that θ parameter is significantly smaller than the β parameter. This is due to the fact that the total population is considered as normalization constant.” For example, in SEIR model dS/dt=-βI S/N-θS, β is divided here by N, i.e. normalized by the total population, which is large number, therefore the ratio between β and θ is high.

2) Concerning the range of parameter values of $\\beta$, in reference [27], they considered the epidemic curves of Italy and Spain assuming $beta$ varies from 1.5 to 3; in this manuscript there are epidemic curves of 61 countries the authors have extended the range from 0 to 4. With the aim of checking the fine tuning, in the revised version the authors have performed a set of experiments varying $\\beta$ from 0 to 10 as well as the range of other parameters, leading to new results exposed in Table 4. Differently from other parameters, in my point of view, the parameters $\\beta$ and $\\theta$ (it is possible to assume a relation between them according to comment 1 above) may vary significantly; since the values are not exposed for all countries, it is not possible to check. It would be nice to know, at least, what is the maximal fitted value of $\\beta$ in order to figure out how significant is the extension of $\\beta$ range of values and if the upper limit was enough to describe all countries.

Answer: The ranges of values of β parameter are shown in box plots with average, maximum and minimum values, as well as outliers in Figure 16. There was an error in naming Figure 16, which was previously said to show relative error values, actually it shows β values. We have expanded the ranged for β further to [0, 8], and the experiments have shown that most of the models use smaller values.

3) The authors pointed out that the optimization problems may lead to unusual situations such as the model with unreported compartment may result to larger errors than the model without it, as pointed out by me and, the extended versions of the models - with progression from susceptible to infected compartments without interaction between them – produces worse results than the usual ones as pointed out by the other reviewer. Since the technical reasons leads to counter-intuitive results, I think it is important to include a sentence about it, maybe in the results section or in the discussion section.

Answer: Thank you for your valuable comments, which were similar to other reviewer. We redesigned the experimental setup and used a more advanced optimization method, L-SHADE-R. We have changed the number of runs, which improves confidence of statistical results, and performed experiments with extended search ranges. Thanks to both reviewers’ suggestions, the experimental setup lead to better results, that are logically consisted, i.e. more model parameters enables better fit. In the new version this was experimentally confirmed (see Table 4 and Figure 15). Therefore, we believe that now there are no counterintuitive results in improved version of the paper.

The fine-tuning phase was omitted in the new experimental procedure, because the new optimization method provided better results, hence there was no point in applying fine-tuning.

Also, an error with parameter representation in Table 6 was fixed, where omega parameter was used instead of alpha.

The updated code and results are available at https://github.com/VladimirStanovov/OWID-COVID-19-Analysis

 

First review response

We would like to thank the reviewers for their insightful suggestions. We have tried our best to address them.

Reviewer #1

1. It is well known that a system of simultaneous nonlinear differential equations is very sensitive in terms of its initial conditions. However, the initial conditions information is missing.

The initial conditions of the models, i.e. state variables, were set according to the available data. First value of the Infected, that were recorded in official statistics, were used for initialization of the Infected state. Number of susceptible was determined according to the population size of each country. In particular, in addition to the mentioned parameters, for every model the algorithm had to tune the ω parameter, responsible for the total number of susceptible at the considered time period. For example, if the total population was 1 million people, setting ω=0.01 meant that the number of susceptible was set to 10000. Other states, such as exposed, recovered and unreported, were initialized to zero.

2. It used a heuristic algorithm to estimate the beginning and the end of the first wave. This algorithm uses some arbitrarily taken parameters. It was not clear if the results are robust by changing these parameters.

Several experiments were conducted to develop and parameterize the proper mechanism to identify first wave. The algorithm was manually checked for all countries. We have added Figure 14 with a short analysis of the influence of τ and wp parameters, controlling the algorithm for determining the first wave.

One should consider that data for some countries were obviously not taken in regular intervals and some irregularities in number of infected cases also occur, like decrease in cumulative number of infected which should not be possible. This was the reason that a general heuristic approach was designed and used for all countries.

The two main parameters controlling the proposed heuristics for determining the first wave are τ and wp. To demonstrate the influence of these values, Figure 14 shows the possible scenarios of determining the first wave with varied parameter values.

3. All extended versions of the well-known SI, SIR, SEIR, and SEUIR models add the new parameter theta. The original SI, SIR, SEIR, and SEUIR models are particular cases of these extended versions when theta goes to zero. So, by construction, the predictive power of the extended versions needs to be at least as good as the original versions (assuming that the same initial condition was used). However, in Table 4, there are some cases when the extended version is worse. This contradiction probably arises from two origins: (i) some limitations of the optimization algorithm itself; (ii) the original and extended versions use different initial conditions. This must be better explained, and a better fitting process must be made.

The optimization algorithm used in the experiments starts with a random set of solutions, and although the search ranges, starting and ending dates are the same, the final results are always different. The search process for standard and modified models is performed in search spaces with different dimensions. Adding more variables results in more local optima.

We have also observed this before the paper submission, however, we considered, that the main information lies in the model structure which has been identified and it’s connected to dynamics and offers certain explanation about the process.

We went further to the fine-tuning of the optimization process. An additional series of experiments has been performed to test if it is possible to improve the results by fine-tuning the standard models. This fine-tuning means that the parameters of the standard model, such as SI, SIR, SEIR or SEIUR are taken as used as a starting point in the population of the DE individuals to search for optimal parameter values of the modified models, the resulting models are marked as BDFT, eSIRFT, eSEIRFT and eSEIURFT. Note that only one of the individuals in DE population was initialized using parameters of standard models, and all other individuals were initialized randomly. The results of these additional experiments have shown that fine-tuning gives either better or at least the same error rates.

4. In the fitting process, the parameters are optimized within some boundaries taken as given. It was not explained why these boundaries are chosen and how robust are the results by changing their values.

The ranges for σ and γ were chosen according to [26], where transmission from exposed to infected and from infected to recovered were determined between 3 and 14 days. For β the search range was set from 0 to 4 according to preliminary experiments and literature, for example in [27] the β parameter was set to from 1.5 to 3.

To estimate whether the ranges of the parameters' search are appropriate, the boxplots were drawn for the β parameter for all considered models. The β parameter was chosen, as it is present in all models, and has significant influence, as it is responsible for new infections caused by contacts between infected and susceptible. Figure 16 shows box plots for all models, the parameter values were taken from all 10 runs on all countries.

 Chu, J. A statistical analysis of the novel coronavirus (COVID-19) in Italy and Spain. PLoS ONE, 2021, 16, https://doi.org/10.1371/journal.pone.0249037

 Schmitt, F.G. An algorithm for the direct estimation of the parameters of the SIR epidemic model from the I(t) dynamics. European Physical Journal plus, 2022, 137, https://doi.org/10.1140/epjp/s13360-021-02237-7

5. The goodness-of-fit is measured using a normalized version of the Euclidean distance between the actual and the predicted cumulative number of cases. It would be interesting to verify the robustness of the results with other metrics.

Preliminary, we carefully examined the fitting process by using several standard measures for fitness function such as Mean Absolute Percent Error (MAPE), Mean Absolute Error (MAE), Mean Squared Error (MSE). However, the proposed relative error measure REc was more stable in case of extreme values e.g. when the number of infected is close to 0.

6. Finally, several works pointed out that COVID-19 epidemics, being a human-to-human driven disease, evolves locally [1]. The infected persons spread the disease to close susceptible persons, not to any suspected person in the entire country. Hence, the model will not work at the national level. A good evolution of your regression analysis is working at some regional level. Additionally, working at this level, the outflow infections can be easily interpreted as internal migration.

[1] Some papers:

Arenas, Alex, et al. "Modeling the spatiotemporal epidemic spreading of COVID-19 and the impact of mobility and social distancing interventions." Physical Review X 10.4 (2020): 041055.

Cardoso, Ben-Hur Francisco, and Sebastián Gonçalves. "Universal scaling law for human-to-human transmission diseases." EPL (Europhysics Letters) 133.5 (2021): 58001.

Abstract models, that were used, consider the possibility of spreading the disease to arbitrary person in the population which is certain limitation of used models. In reality, the infected persons spread the disease to close susceptible persons.

Therefore, one should consider, that human-to-human driven diseases, evolves locally. For better fitting results spatiotemporal spreading of infections should be considered, however, with the available limited dataset, the development of more detailed model could not be performed. This is also the reason that most predictions were based on abstract models such as the ones used in our study.

Development of spatiotemporal models should be addressed in the future with the usage of new technologies, that enable the tracking and monitoring of health data of everyone. This on the other hand raises the question of privacy just to mention one.

The main contribution of the paper lies in the identification of possible new mechanisms that underlies the Covid-19 spread by examining different model structures.

In the discussion section, the following part was added: “One should be aware, that the spread of infectious disease inevitably involves spatially connected dynamics, which could be modelled in more details. Several studies pointed out that COVID-19 epidemics, being a human-to-human driven disease, evolves locally [29, 30]. The infected persons spread the disease to close susceptible persons, not to any suspected person in the entire country. One should note, that this is the limitation of the presented models, which might be used on an abstract, aggregate level.”

 

Reviewer #2

1) Concerning the original Bass diffusion model, it is possible to become infected by external sources. Is it reasonable for COVID-19 transmission? Nowadays it is assumed that its transmission is mainly by infected people; the risk of transmission by contaminated surfaces is considered very low. I suggest the authors present some references concerning that discussion, from the epidemiological point of view since it is a central point of the manuscript.

We added references to the following part of the discussion: “This would suggest that a significant part of the dynamics is not due to the contact between the susceptible and infected [31], but because of some external factors [32].“

The following part has been added in the discussion: “According to [33], “The ability of SARS-CoV-2 to remain viable longer on surfaces taken together with its higher virulence in establishing an infection makes it very likely that this coronavirus uses other modes of transmission in addition to respiratory droplets”. This could be one explanation of the observed higher efficiency of the modified models.“

 Jayaweera, M., Perera, H.R., Gunawardana, B., & Manatunge, J. (2020). Transmission of COVID-19 virus by droplets and aerosols: A critical review on the unresolved dichotomy. Environmental Research, 188, 109819 – 109819, https://doi.org/10.1016/j.envres.2020.109819

 Lewis, D. (2021). COVID-19 rarely spreads through surfaces. So why are we still deep cleaning? Nature, 590 7844, 26-28, https://doi.org/10.1038/d41586-021-00251-4

 Galbadage, T., Peterson, B.M., & Gunasekera, R.S. (2020). Does COVID-19 Spread Through Droplets Alone? Frontiers in Public Health, 8, https://doi.org/10.3389/fpubh.2020.00163

2) Assuming the hypothesis that infection may occurs without contact with infected people, your results show that the modified models, inspired in Bass diffusion model, fit data better than the original ones. My other main question is related to the parametrization: beside the data and the fitting methods, the parameters obey ranges of values related to the dynamics of the disease. May the authors justify the parameters’ ranges of values used in the simulations for the set of considered countries? For instance, $\\beta$ varies from 0 to 4 and $\\theta$ varies from 0 to 0.1, but in the discussion the authors conclude that “significant part of the infection is due to external sources. May the authors discuss this point?

The ranges for σ and γ were chosen according to preliminary experiments and literature [26], where transmission from exposed to infected and from infected to recovered were determined between 3 and 14 days. For β the search range was set from 0 to 4 according to preliminary experiments and literature, for example in [27] the β parameter was set to from 1.5 to 3.

To estimate whether the ranges of the parameters search are appropriate, the boxplots were drawn for the β parameter for all considered models. The β parameter was chosen, as it is present in all models, and has significant influence, as it is responsible for new infections caused by contacts between infected and susceptible. Figure 16 shows box plots for all models, the parameter values were taken from all 10 runs on all countries.

 Chu, J. A statistical analysis of the novel coronavirus (COVID-19) in Italy and Spain. PLoS ONE, 2021, 16, https://doi.org/10.1371/journal.pone.0249037

 Schmitt, F.G. An algorithm for the direct estimation of the parameters of the SIR epidemic model from the I(t) dynamics. European Physical Journal plus, 2022, 137, https://doi.org/10.1140/epjp/s13360-021-02237-7

3) In relation to figures 7, 8, 9 and 10, where the usual models are compared to their modified versions, wasn’t it possible to reach the same curve of the modified versions changing the parameter values of the usual models? If it is possible, what are the arguments not to do that procedure? For instance, in Figure 9, changing the value of $\\beta$ of SIR model, the curve of eSIR may be reached.

For particular cases, we could tune the parameters of the standard model to closely fit to the modified model, but it is not possible to get equal response, especially for more complex models.

This could also be examined by analytical solutions which, for the simple cases, might be obtained. The analytical solutions are different thus it is not possible to get exactly the same response, however close fit could be possible.

4) The authors say that they eliminate countries as Denmark, Brazil, Uganda, Vietnam, Poland and Northern Macedonia from their analysis “due to peculiarities in the data”. It is supposed that those peculiarities are different for the cited countries. However, at least in the case of Brazil, it would be relevant to explain its exclusion since it was the second country in the number of reported cases during the first-wave of COVID-19, and for part of the period, it was the first country in the number of deaths due to the disease.

Brazil data did not exercise the clear border between first and second wave same as some other listed countries. As long as our goal was to work with the first wave it was not suitable for the analysis. The data of mentioned countries is still valid though not suitable to examine the dynamics of the first wave, which should be clearly present in data.

5) Looking at the errors of the results for 61 countries, in table 3, for different models, I think that the authors may include, beside the table, a graphical representation of the errors in order to provide a better visualization for comparison between the countries. Due to the number of countries (61) and the number of models (8), the graphics of errors may be presented in a supplementary material.

We have added Figure 15, where the SEIR model is chosen as baseline, and other models’ relative errors are scaled. Also, the countries are ordered for better visibility.

6) It was curious that the model with unreported cases do not provide good results. Is it associated to the number of parameters or to its variability. Do the authors have any justification for that result?

In order to compensate the complexity of optimization problems in cases like SEIUR model, the number of generations of DE was increased to 600. According to Figure 15, the models with unreported cases perform similarly to the SEIR and eSEIR models, although a little larger error values were observed. Our justification is that there are several local optima in the search space, and the optimization process converges to one of them. These local optima generate similar dynamics, each providing close fit with the data. The complexity of models with unreported cases adds to the variability, and significantly increases the number of possible local optima, whereas for models without it the optimization converges faster.

7) I suggest the authors include the analytical expressions for the basic reproduction number for all models in order to make the comparison easier for the reader.

We have provided the reproduction number R0=β/γ only for the SIR model based on the estimated parameters. Determination of R0 number for SEIR and SEIUR models is challenging according to several studies (Heng et al., 2020) and it was not in the main focus of our study. For SEIR and SEIUR models there are analytical solutions, however, they require the death rate to be considered, which was not the case in our study. If the death rate is set to 0, these equations simplify to R0=β/γ, however, as β and γ are in different equations, the results of such calculation are often unstable, i.e. R0 could be as large as 30, or as small as 0.3.

Heng, K., & Althaus, C.L. (2020). The approximately universal shapes of epidemic curves in the Susceptible-Exposed-Infectious-Recovered (SEIR) model. Scientific Reports, 10.

8) I recommend that the authors include in caption of figure 6 the fitted values of parameters that lead to the curves exhibited in figure 6. It may help the reproduction of the results

We have added Table 7 with all parameter values to reproduce the graphs in Figure 17 (the figure numbers were changed). Added: “The values in Tables 7 and 8, combined with the description of all models can be used to reproduce the performed experiments.”

---

## [Decision Letter · Decision Letter 3]

7 Dec 2022

Identification of COVID-19 Spread Mechanisms Based on First-Wave Data, Simulation Models, and Evolutionary Algorithms

PONE-D-22-12244R3

Dear Dr. Stanovov,

We’re pleased to inform you that your manuscript has been judged scientifically suitable for publication and will be formally accepted for publication once it meets all outstanding technical requirements.

Kind regards,

Sebastián Gonçalves, Ph.D.

Academic Editor

PLOS ONE

Additional Editor Comments (optional):

Reviewers' comments:

Reviewer's Responses to Questions

**Comments to the Author**

1. If the authors have adequately addressed your comments raised in a previous round of review and you feel that this manuscript is now acceptable for publication, you may indicate that here to bypass the “Comments to the Author” section, enter your conflict of interest statement in the “Confidential to Editor” section, and submit your "Accept" recommendation.

Reviewer #2: All comments have been addressed

2. Is the manuscript technically sound, and do the data support the conclusions?

Reviewer #2: Yes

3. Has the statistical analysis been performed appropriately and rigorously? 

Reviewer #2: Yes

4. Have the authors made all data underlying the findings in their manuscript fully available?

Reviewer #2: Yes

5. Is the manuscript presented in an intelligible fashion and written in standard English?

Reviewer #2: Yes

6. Review Comments to the Author

Reviewer #2: After this last revision of the manuscript, Iam in favor of its publication in PLOS ONE.

The authors answered satisfactory the raised points by me

7. PLOS authors have the option to publish the peer review history of their article (what does this mean?). If published, this will include your full peer review and any attached files.

Reviewer #2: No

---

## [Editor Report · Acceptance letter]

16 Dec 2022

PONE-D-22-12244R3 

Identification of COVID-19 Spread Mechanisms Based on First-Wave Data, Simulation Models, and Evolutionary Algorithms 

Dear Dr. Stanovov:

I'm pleased to inform you that your manuscript has been deemed suitable for publication in PLOS ONE. Congratulations! Your manuscript is now with our production department. 

Kind regards, 

on behalf of

Dr. Sebastián Gonçalves 

Academic Editor

PLOS ONE